# Learning and Retrieval from Prior Data for Skill-based Imitation Learning

**Soroush Nasiriany[1], Tian Gao[1,2], Ajay Mandlekar[3], Yuke Zhu[1]**

[1]The University of Texas at Austin, [2]IIIS, Tsinghua, [3]NVIDIA Research

**Abstract:** Imitation learning offers a promising path for robots to learn general-purpose behaviors, but traditionally has exhibited limited scalability due to high data supervision requirements and brittle generalization. Inspired by recent advances in multi-task imitation learning, we investigate the use of prior data from previous tasks to facilitate learning novel tasks in a robust, data-efficient manner. To make effective use of the prior data, the robot must internalize knowledge from past experiences and contextualize this knowledge in novel tasks. To that end, we develop a skill-based imitation learning framework that extracts temporally extended sensorimotor skills from prior data and subsequently learns a policy for the target task that invokes these learned skills. We identify several key design choices that significantly improve performance on novel tasks, namely representation learning objectives to enable more predictable skill representations and a retrieval-based data augmentation mechanism to increase the scope of supervision for policy training. On a collection of simulated and real-world manipulation domains, we demonstrate that our method significantly outperforms existing imitation learning and offline reinforcement learning approaches. Videos and code are available at `https://ut-austin-rpl.github.io/sailor`

**Keywords:** Imitation Learning, Skill Learning, Robot Manipulation

## 1 Introduction

A long-standing dream in robotics is to enable robots to perform general-purpose tasks in diverse environments. In recent years, imitation learning has led to great progress towards this goal. Recent work has demonstrated that well-engineered behavioral cloning methods can achieve competitive performance on diverse manipulation tasks [1, 2, 3]. Despite this promise, these imitation learning algorithms have been confined to relatively small-scale domains. For real-world problems, current imitation learning algorithms often demand a large number of task demonstrations, which can be difficult and costly to obtain.

Realizing these limitations, a series of recent work seeks to use prior interaction data to improve the sample efficiency of imitation learning on new tasks. Such prior data come in various forms, including task-agnostic exploratory "play" data [4] or demonstrations previously collected for different tasks [5]. An open question is how best to extract knowledge from these large prior datasets and use this knowledge to facilitate learning novel tasks. One promising approach is skill-based imitation learning [6, 7], which aims to learn a latent space of short-horizon sub-trajectories from the prior data (called *skill learning*) and subsequently learn a policy to invoke the skills to solve a specific downstream task (called *policy learning*). This approach offers several appealing advantages: First, the policy benefits from the temporal abstraction encapsulated by the skills, allowing the policy to focus on higher-level reasoning about *what* behavior to perform rather than *how* to execute that behavior; and second, by reasoning about the target task in relation to skills trained on prior data, the robot implicitly distills knowledge from the rich diverse interactions of the prior data into the policy. Even so, existing skill-based imitation learning approaches bring modest improvements over simple behavioral cloning methods. This work aims at identifying the underlying limitations of existing approaches and designing new methods to address these limitations.

6th Conference on Robot Learning (CoRL 2022), Auckland, New Zealand.

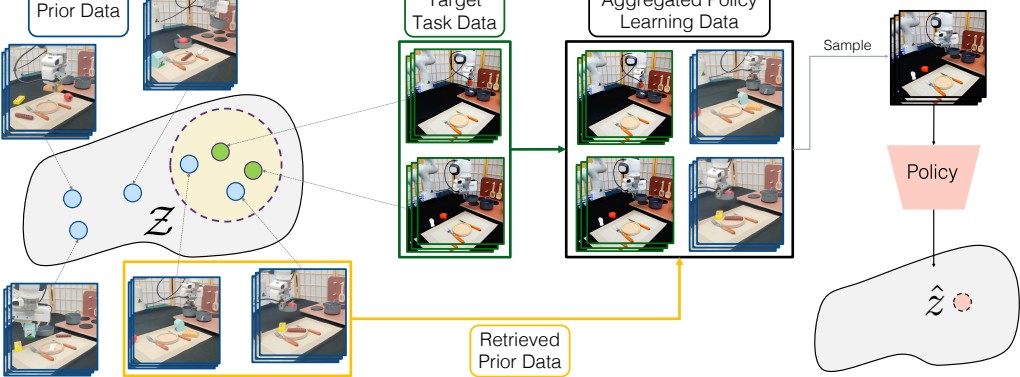

1. Skill representation learning    2. Retrieval for policy learning    3. Policy learning

Figure 1: **Overview.** We present a skill-based imitation learning framework that uses prior data to effectively learn novel tasks. First, we learn a latent skill model on the prior data, with objectives to ensure a predictable skill representation. Given target task demonstrations, we use this latent space to retrieve similar behaviors from the prior data, expanding supervision for the policy. We then train a policy which outputs latent skills.

We argue that a practical approach should encompass two key properties. First, the latent skill space should serve as a *predictable* representation for downstream policy learning, allowing the policy to accurately infer which appropriate skill to use in new situations. While prior methods [8, 9, 6, 7] have widely adopted variational autoencoders to transform sub-trajectories into embeddings, we empirically show that this learning objective alone is insufficient for constructing a well structured latent space. As a result, the policy tends to execute irrelevant skills. Second, the robot should take advantage of the prior data to learn both the skills and the policy. A major limitation of existing skill-based imitation learning approaches [6, 7] is that they primarily focus on using prior data for skill learning but not policy learning. In these approaches, policies learned on a small number of target task demonstrations are prone to severe overfitting and covariate shift.

To address these limitations, we introduce **S**kill-**A**ugmented **I**mitation **L**earning with pri**o**r **R**etrieval (SAILOR). To improve the predictability of our skill representations, SAILOR uses an auxiliary temporal predictability objective to estimate the temporal distance between sub-trajectories in the demonstration sequences. To use prior data for policy learning, we introduce a novel retrieval-based data augmentation procedure that selectively retrieves data relevant to the target task. Specifically, we consider a subset of sub-trajectories from the prior dataset that have high latent skill similarity with sub-trajectories in the target task demonstrations (see Fig. 1). We evaluate on a wide variety of manipulation tasks in simulation and the real world, and show that SAILOR significantly outperforms state-of-the-art imitation learning and offline reinforcement learning approaches. Through comprehensive analysis, we highlight the roles that prior data, our representation learning objective, and retrieval-based data augmentation have in data-efficient learning of robust manipulation policies.

## 2 Related Work

**Learning from Prior Data.** There is a large body of work on learning manipulation tasks using human demonstrations [1, 10, 11, 12, 13, 14]. While promising, most of these works learn tasks independently, without re-using knowledge from prior tasks. As a result, they have high data requirements and exhibit brittle generalization in complex long-horizon tasks. To address these limitations, several lines of work have investigated leveraging large offline prior datasets to facilitate learning downstream robotic tasks. These prior datasets include task-agnostic play data [4, 15], demonstrations for related tasks [5], self-supervised agent-generated data [16], or a combination of these [17]. Alternatively to these, large video datasets are an appealing choice [18, 19, 20]. Recent work has leveraged these datasets to learn pre-trained visual representations for downstream control tasks [21, 22]. Yet despite their appeal, after experimenting with one such approach, R3M [21], we found that it can sometimes hinder downstream performance. One hypothesis is that the prior data on which these methods are trained on exhibit significant domain shift compared to downstream tasks, limiting transfer. In this work, we instead consider large multi-task robotic datasets.

**Multi-Task Imitation Learning.** Multi-task imitation learning methods offer a promising way to learn from diverse robotic data. These include task-conditioned [5], language-conditioned [23, 4, 24, 25], and skill-based [6, 7] imitation learning. While most task-conditioned and language-conditioned approaches seek to learn a single policy that performs well across a series of tasks, we focus on learning task-specific policies by utilizing a large multi-task prior dataset to learn a useful skill representation space, and supervising the task-specific policy using learned latent skills. These multi-task imitation approaches could be complementary to our skill representation learning pipeline, ie., we can consider incorporating language and task ID supervision into our approach.

**Skill-based Imitation Learning.** We adopt skill-based imitation learning as the underlying framework for our method. The objective is to learn temporally abstract representations of sensory-motor data (termed *skills*) to enable more effective imitation learning. A long line of work has proposed learning skill representations by segmenting demonstrations into trajectory segments termed *sub-trajectories*. Among these include work on segmenting demonstrations into variable-length sub-trajectories, either in an unsupervised manner [26, 27, 28, 29, 30, 31, 32] or relying on additional supervision [33, 34, 35, 36]. Recent work has shown promise for encoding fixed length sub-trajectories without any additional supervision [8, 9, 6, 7] using variational autoencoder-based approaches. We adopt this setting due its relative simplicity and scalability. In contrast to these prior approaches (see Sec. 3), we enforce two key properties—a temporal predictability objective in our learned skill representation and a retrieval-based mechanism to improve task-specific policy learning.

## 3 Problem Formulation

Our goal is to leverage prior data to effectively learn novel target tasks in a data-efficient manner. Formally, we consider a target task as a Markov Decision Process $\mathcal{M} = (\mathcal{S}, \mathcal{A}, r, p, p_0, \gamma)$ representing the state space, action space, reward function, transition probability, initial state distribution, and discount factor. Our objective is to learn a policy that maximizes the discounted sum of rewards for the task. To learn the policy, we assume access to a small offline dataset $\mathcal{D}_{\text{target}}$ collected for the target task and a large offline dataset $\mathcal{D}_{\text{prior}}$ either from previous related tasks or task-agnostic interactions. These datasets consist of variable-length trajectories in the form $\{o_0, a_0, o_1, \cdots, o_{T_i}\}$ with $o_i \in \mathcal{O}$ denoting the observations and $a_i \in \mathcal{A}$ denoting actions. We highlight that $\mathcal{D}_{\text{prior}}$ and $\mathcal{D}_{\text{target}}$ may have significant differences, i.e., the two datasets may come from different environments, be collected by different human demonstrators, and demonstrate different tasks.

**Skill-based Imitation Learning.** We employ a skill-based imitation learning framework [6, 7] which consists of two stages: skill learning and policy learning. In the skill learning phase, we learn a skill embedding space $\mathcal{Z} \subset \mathbb{R}^d$ of fixed-length sub-trajectories $\tau = \{o_0, a_0, o_1, \cdots, o_{H-1}, a_{H-1}, o_H\}$ in $\mathcal{D}_{\text{prior}}$. These skill embeddings serve as an abstract representation of the agent's behavior and can be invoked to solve a range of downstream tasks. A number of representation learning methods can be employed to learn the skill embeddings, spanning reconstruction-based methods and contrastive learning. In the subsequent policy learning phase, we are given $\mathcal{D}_{\text{target}}$ and our goal is to learn a policy $\pi$ for the target task $\mathcal{T}$. The policy now emits skill embeddings $z \in \mathcal{Z}$ and during execution we decode $z$ into a sequence of actions $\{a_0, a_1, \cdots, a_{H-1}\}$ via a skill decoder model $p_\psi$. To learn the policy, prior work has proposed parametric [6] and semi-parametric [7] polices that first parse segments of $\mathcal{D}_{\text{target}}$ into skills and subsequently learn to map observations in $\mathcal{D}_{\text{target}}$ to these skill embeddings. Note that we use distinct terms—*skill* and *policy*—to distinguish the role of these two components. Skills represent short-horizon behaviors that can be re-used across many tasks, while the policy solves a specific long-horizon target task.

## 4 Skill-based Imitation Learning with Retrieval

In this section, we describe our skill-based imitation learning approach that can leverage prior multi-task data to efficiently learn novel target tasks with a small amount of task-specific demonstrations. As discussed in Sec. 3, this consists of two phases—a task-agnostic skill learning phase, where a latent skill space is learned using the prior data, and a task-specific policy learning phase where task-specific data is used to learn a policy using the skills as supervision. Compared to prior methods, our approach makes two key considerations—we (1) ensure that the learned skill space is a *predictable*

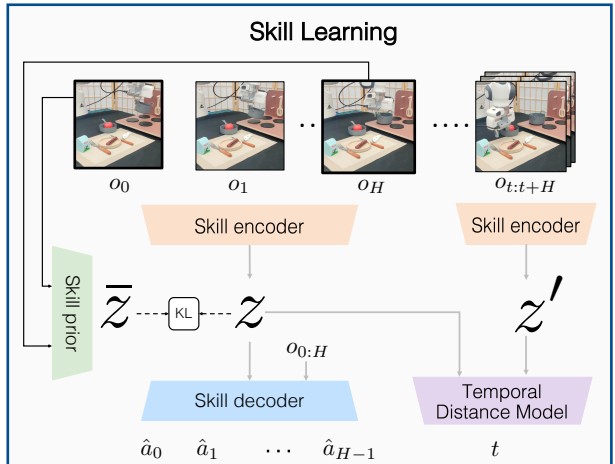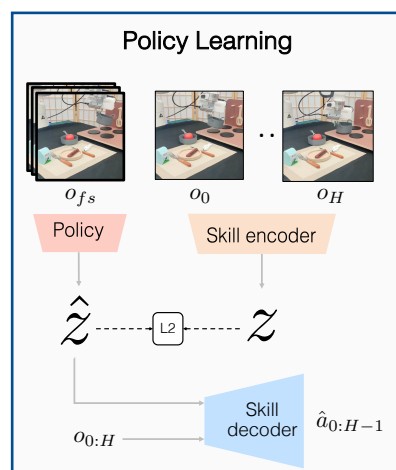

Figure 2: **Model Overview.** Our method consists of a skill learning and policy learning phase. (Left) In the skill learning phase, we learn a latent skill representation of sub-trajectories via a variational autoencoder. We include an additional temporal predictability term to learn a more consistent latent representation. (Right) In the policy learning phase, we train the policy to predict the latent skill given a history of observations preceding the sub-trajectory. To execute the policy, we decode the predicted latent using the skill decoder.

representation for downstream policy learning, and (2) improve the efficacy of task-specific policy learning by *retrieving* task-relevant datapoints from the prior dataset. See Fig. 2 for our model overview.

### 4.1 Learning a Predictable Representation of Skills

We learn a skill representation by encoding sub-trajectories with a variational autoencoder (VAE) [37], and we further introduce an auxiliary objective to shape the representation. Denoting a given sub-trajectory as $\tau$, we employ a long short-term memory (LSTM) [38] encoder $q_\phi$ that encodes $\tau$ into a Gaussian distribution over latent skills. Our decoder is an LSTM network $p_\psi$ that, for each timestep $t$, decodes a latent $z$ and the given observation $o_t$ into the reconstructed action $\hat{a}_t$. We additionally employ a learned prior $p_\theta$ to encourage sub-trajectories with similar starting and ending observations to have similar latent representations [15]. Our VAE loss objective is then

$$\mathcal{L}_{\text{VAE}}(\phi, \psi, \theta) = -\mathbb{E}_{z \sim q_\phi(z|\tau)} \left[ \sum_{t=0}^{H-1} \log p_\psi(a_t|z, o_t) \right] + \beta \cdot D_{KL}(q_\phi(z|\tau) || p_\theta(z|o_0, o_H)), \quad (1)$$

where $\beta$ controls the effect of the KL divergence term [39][1].

It is important to highlight that action reconstruction is not the sole objective of our skill learning model—learning a consistent and predictable representation of behavior is critical for downstream policy learning, as shown by recent work [40, 41]. While the KL divergence term is one step towards this objective (by encouraging skills to be predictable given partial information from the sub-trajectories), in this work we introduce an additional *temporal predictability* term that encourages the learned latent space to predict the temporal difference between two sub-trajectories. Specifically, given two-sub-trajectories $\tau_1$ and $\tau_2$ from the same trajectory separated by $t$ timesteps, we learn a model $m_\omega$ to predict $t$ given the corresponding skill mean embeddings of the trajectories:

$$\mathcal{L}_{\text{TP}}(\omega, \phi) = \left( m_\omega \left( \mu(q_\phi(z|\tau_1)), \mu(q_\phi(z|\tau_2)) \right) - t \right)^2, \quad (2)$$

where $\mu$ denotes taking the mean of the distribution. We back-propagate $\mathcal{L}_{\text{TP}}$ through the skill encoder model, allowing the this term to shape the learned skill representation. Note that this is just one way to encourage temporal predictability, other objectives are also readily compatible with our method, such as time-contrastive networks [42]. Our overall objective is a weighted combination of the VAE and temporal predictability objectives:

$$\mathcal{L}_{\text{Skill}}(\phi, \psi, \theta, \omega) = \mathcal{L}_{\text{VAE}}(\phi, \psi, \theta) + \alpha \mathcal{L}_{\text{TP}}(\omega, \phi). \quad (3)$$

---

[1]In practice we use a deterministic VAE decoder and we compute the reconstruction loss using $\ell_2$ distance.

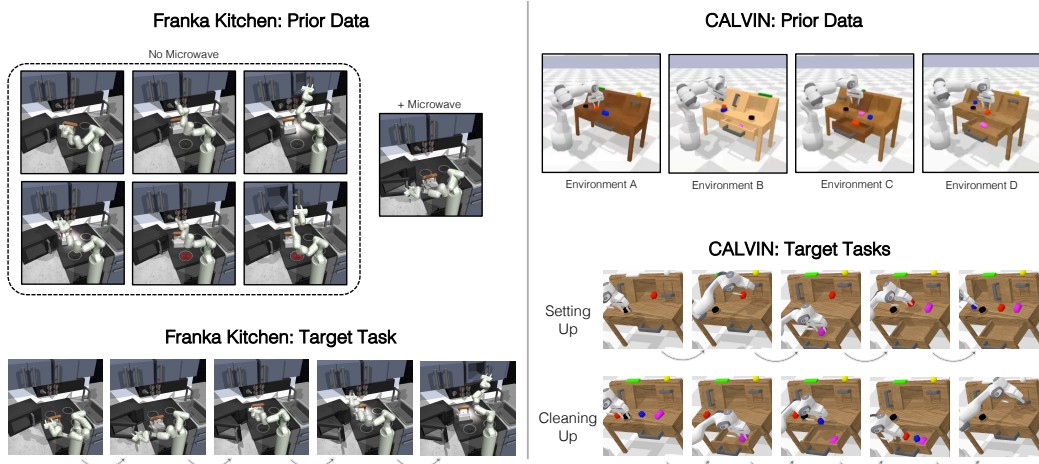

Figure 3: **Simulated Tasks.** We perform extensive evaluations on two simulation domains. (Left) Franka Kitchen: our target task involves a specific permutation of four subtasks and we consider two prior datasets: demonstrations involving all subtasks and demonstrations involving all subtasks except opening the microwave. (Right) CALVIN: we adopt the play dataset of Mees et al. [4] as our prior data and perform evaluations on two target tasks: setting up the playroom environment and, conversely, cleaning up the environment.

Refer to Algorithm 1 for a detailed summary of our skill learning algorithm.

## 4.2 Retrieval-based Policy Learning

In the policy learning phase, we employ an LSTM policy that outputs the skill $z$ to execute next. We train the policy on a dataset $\mathcal{D}_{\text{policy}} = \{(o^i_{fs}, z_i = \mu(q_\phi(\tau_i)))\}$, where $\tau_i$ is an $H$-length sub-trajectory, $z_i$ is the mean encoding of that sub-trajectory, and $o^i_{fs}$ is the frame-stacked history of $F$ observations preceding the sub-trajectory. We train the policy to predict $z_i$ from $o^i_{fs}$ using a standard behavioral cloning loss. During execution, we roll out the LSTM skill decoder $p_\psi$ in a closed-loop manner, i.e., at each timestep we observe $o_t$ and execute the next action $a_t = \mu(p_\psi(z, o_t))$. After rolling out the skill for $H$ timesteps we repeat the process by sampling a new skill from the policy.

A common approach to skill-based policy learning is training the policy with all $H$-length sub-trajectories in $\mathcal{D}_{\text{target}}$. However, this limits the amount of supervision, especially when $\mathcal{D}_{\text{target}}$ is small. On the other hand, naïvely training on all sub-trajectories in $\mathcal{D}_{\text{target}}$ and $\mathcal{D}_{\text{prior}}$ can hurt performance [43] due to divergent and conflicting behaviors between the prior and target datasets. We thus introduce a retrieval-based mechanism to train on sub-trajectories in $\mathcal{D}_{\text{prior}}$ that have high similarity with those in $\mathcal{D}_{\text{target}}$. While many similarity metrics are suitable, in this work we measure similarity with respect to the skill embedding space—intuitively, sub-trajectories with similar skill embeddings demonstrate similar behaviors. First we obtain skill embeddings of randomly sampled sub-trajectories in $\mathcal{D}_{\text{prior}}$ and $\mathcal{D}_{\text{target}}$:

$$Z_{\text{prior}} = \{\mu(q_\phi(\tau_i))\}, \tau_i \sim \mathcal{D}_{\text{prior}}; \qquad Z_{\text{target}} = \{\mu(q_\phi(\tau_j))\}, \tau_j \sim \mathcal{D}_{\text{target}}. \qquad (4)$$

We then calculate the pairwise $\ell_2$ distances between the prior and target dataset skill embeddings, i.e., `D[i][j]` $= \|Z^i_{\text{prior}} - Z^j_{\text{target}}\|_2$. Next, for each prior dataset skill embedding $z_i \in Z_{\text{prior}}$, we find the closest corresponding target dataset skill, `D_min[i] = min(D[i][:])`. Finally, we retrieve the top-$n$ sub-trajectories in $\mathcal{D}_{\text{prior}}$ with the smallest distance `argsort(D_min)[:n]`, resulting in the retrieval dataset $\mathcal{D}_{\text{ret}}$. We train the policy using the aggregated set of sub-trajectories in $\mathcal{D}_{\text{ret}}$ and $\mathcal{D}_{\text{target}}$. We use a behavioral cloning loss split across two terms: one for $\mathcal{D}_{\text{target}}$, and one for $\mathcal{D}_{\text{ret}}$ weighted by a factor $\gamma$ to control the effect of the retrieval data relative to the target dataset. At the same time as training the policy we additionally fine-tune the skill model on $\mathcal{D}_{\text{target}}$. We summarize the retrieval, policy learning, and skill fine-tuning steps in Algorithm 2.

| Dataset | BC-RNN | BC-RNN (FT) | BC-RNN (R3M) | IQL | IQL (UDS) | FIST | SAILOR (ours) |
|---|---|---|---|---|---|---|---|
| Kitchen-All | $32.3 \pm 9.0$ | $93.0 \pm 2.3$ | $83.3 \pm 2.1$ | $65.0 \pm 7.1$ | $78.3 \pm 7.0$ | $68.7 \pm 2.5$ | $\mathbf{95.7 \pm 1.7}$ |
| Kitchen-No Microwave | | $77.7 \pm 7.6$ | | | $70.3 \pm 7.3$ | $83.0 \pm 0.8$ | $\mathbf{95.0 \pm 2.2}$ |
| CALVIN-Setting Up | $33.0 \pm 2.2$ | $72.8 \pm 5.4$ | $26.0 \pm 1.4$ | $7.7 \pm 1.7$ | $16.3 \pm 2.6$ | $3.7 \pm 2.4$ | $\mathbf{77.3 \pm 3.1}$ |
| CALVIN-Cleaning Up | $41.0 \pm 1.6$ | $61.8 \pm 9.1$ | $28.0 \pm 2.2$ | $14.7 \pm 2.1$ | $20.0 \pm 3.7$ | $9.3 \pm 5.0$ | $\mathbf{88.0 \pm 5.1}$ |

Table 1: **Quantitative evaluation on two simulation domains.** We evaluate our method SAILOR against a set of six baselines and report the mean task success rate and standard deviation over three seeds (exception: six seeds for BC-RNN (FT) due to high variance). Note: for the kitchen tasks we report one number for baselines that do not involve prior data. We see that SAILOR significantly outperforms the baselines on all tasks.

# 5 Experiments

## 5.1 Simulated Experiment Setup

We perform empirical evaluations on two simulated robot manipulation domains (see Fig. 3):

**Franka Kitchen** [44]: A simulated kitchen environment involving different sub-tasks, such as opening cabinets, moving a kettle, and turning on a stove. This environment comes with a large dataset of approximately 600 demonstrations performing various permutations of seven subtasks. In this dataset, a subset of 18 demonstrations correspond to $\mathcal{D}_{\text{target}}$ and demonstrate a specific permutation of subtasks: opening the microwave, followed by moving the kettle, flipping on the light switch, and opening the sliding cabinet. We consider two prior datasets $\mathcal{D}_{\text{prior}}$: (1) using all demonstrations except the ones corresponding to the target task (`Kitchen-All`); and (2) using all demonstrations except those that involve interacting with the microwave (`Kitchen-No Microwave`). These prior datasets have 584 and 235 demonstrations, respectively.

**CALVIN** [4]: A simulated tabletop playroom environment accompanies by a large dataset of task-agnostic "play" data with 2.3M transitions. The play data encompass diverse behaviors, such as opening and closing drawers, turning on and off the lights, and picking, placing, and pushing blocks. We use all play data as $\mathcal{D}_{\text{prior}}$ to solve two target tasks. The first target task involves setting up the playroom environment in multiple stages (`CALVIN-Setting Up`). Specifically, the robot must turn on the lights, and retrieve three blocks and place them on the table. The second target task in contrast involves cleaning up the playroom environment (`CALVIN-Cleaning Up`). Specifically, the robot must open the drawer, place all three blocks into the drawer, close the drawer, and turn off the lights. For each task, we collect 30 demonstrations, which amounts to about half an hour of data collection.

The CALVIN domain is substantially more challenging than the Franka Kitchen domain, as the target tasks have a longer horizon and involve a greater number of objects. Also, in contrast to Franka Kitchen, the prior and target datasets in CALVIN are collected by *different human demonstrators* who exhibit different styles of teleoperation.

We learn vision-based policies for both domains. We refer readers to Appendix B for a detailed discussion of our tasks and datasets, and Appendix C.5 for environment implementation details.

## 5.2 Quantitative Analysis

We evaluate our method SAILOR against state-of-the-art imitation learning and offline reinforcement learning algorithms:

**BC-RNN**: behavioral cloning on $\mathcal{D}_{\text{target}}$ without prior data. We adopt the LSTM-based BC-RNN implementation in robomimic [1], which has shown superior performance over other behavioral cloning approaches.

**BC-RNN (FT)**: BC-RNN variant that leverages prior data. We first pre-train BC-RNN on $\mathcal{D}_{\text{prior}}$ and subsequently fine-tune on $\mathcal{D}_{\text{target}}$. This baseline aims to examine the effectiveness of supervised pre-training on interaction data for imitation learning.

**BC-RNN (R3M)**: behavioral cloning on $\mathcal{D}_{\text{target}}$ using a frozen R3M visual representation [21] pre-trained on the large-scale Ego4D video dataset [20]. This baseline intends to examine the effectiveness of using visual representations trained on natural images and videos.

| Dataset | Ours | No TP | No Retrieval | All Retrieval | No Prior Data |
|---|---|---|---|---|---|
| CALVIN-Setting Up | $\mathbf{77.3 \pm 3.1}$ | $68.0 \pm 3.7$ | $65.0 \pm 2.2$ | $64.3 \pm 10.8$ | $50.7 \pm 6.5$ |
| CALVIN-Cleaning Up | $\mathbf{88.0 \pm 5.1}$ | $74.7 \pm 7.9$ | $70.0 \pm 0.8$ | $65.3 \pm 7.3$ | $60.7 \pm 3.1$ |

Table 2: **Ablation Results.** We find that temporal predictability and retrieval are critical to skill-based imitation learning, as without these components the agent struggles against a naive BC-RNN (FT) baseline. In addition, we validate that prior data plays a large role in the performance of our method.

**IQL**: Implicit Q-Learning [45], a recent offline reinforcement learning method with state-of-the-art performance on the D4RL dataset [46]; trained on $\mathcal{D}_{\text{target}}$.

**IQL (UDS)**: Implicit Q-Learning with Unlabeled Data Sharing [47, 48], which is a variant of IQL trained jointly on $\mathcal{D}_{\text{prior}}$ and $\mathcal{D}_{\text{target}}$, where the transitions in $\mathcal{D}_{\text{prior}}$ are labeled with the minimum reward (0 for our tasks). Singh et al. [47] and Yu et al. [48] show that this simple data augmentation procedure can effectively leverage prior data without additional rewards annotation.

**FIST**: Few-shot Imitation with Skill Transition Models [7], an analogue of our method that employs a semi-parametric policy to select the skill to execute next. This baseline uses the same underlying skill model as our method but a different policy learning scheme and does not involve retrieval.

Refer to Appendix C for additional implementation details on our model architecture, pseudocode, and evaluation protocols, and Appendix D for hyperparameter details. We report performance of all methods in Table 1. SAILOR greatly outperforms the baselines with an average task success rate of 89.0%. Notably it outperforms the most competitive baseline BC-RNN (FT) by 12.7%. BC-RNN performs poorly as it fails to learn an effective policy from a small number of demonstrations. In comparison, BC-RNN (R3M) shows significant improvements on the Franka Kitchen tasks, but performs worse on the CALVIN tasks. We hypothesize that this is due to the limited generalization ability of the pre-trained visual representations. The offline reinforcement learning baselines show more promising results on the Franka kitchen tasks but struggle on the more challenging CALVIN tasks. Finally, FIST significantly under-performs our method. As FIST uses the same underlying skill model as our method, we attribute the limitations of FIST to its semi-parametric policy.

## 5.3 Ablation Study

We perform an extensive ablation study to understand the effects of various modeling choices on our method. First, we study the effect of the temporal predictability term in Eq. (2) on downstream task performance by removing it from Eq. (3) (No TP). Next we study the role of retrieval by training the policy solely on sub-trajectories in $\mathcal{D}_{\text{target}}$ (No Retrieval). We also study the opposite case— training the policy on all of $\mathcal{D}_{\text{prior}}$ and $\mathcal{D}_{\text{target}}$ (All Retrieval). Finally, we study the role of prior data on our method by training the skill and policy solely on $\mathcal{D}_{\text{target}}$ (No Prior Data). We also report ablations in Appendix A on the size of the prior, retrieval, and target datasets, in addition to the choice of retrieval method.

We present results in Table 2 for the more challenging CALVIN tasks. First, we find that both the temporal predictability objective and the retrieval mechanism have a significant impact on the final performance. It is worth noting that removing these components makes our model degenerate into the skill-based imitation learning setting of OPAL [6]. In fact, these two ablations perform worse than the naïve BC-RNN (FT) baseline for the CALVIN-Cleaning Up task, indicating that both an effective skill representation and the retrieval mechanism play a critical role in skill-based imitation learning. The All Retrieval ablation also performs suboptimally—qualitatively we observe that the robot often was "distracted" and performed behaviors unrelated to the target task, likely due to the multimodal distributions in the prior data. Finally, the No Prior Data ablation validates the role of prior data in learning effective skills and policy. Comparing the No Retrieval ablation and the No Prior Data ablation, we see a $10 - 15\%$ gap in performance, and this is attributed to the fact that the skills in the No Retrieval ablation are additionally trained on the prior data. Despite the loss of performance in the No Prior Data ablation, we still see that it outperforms the BC-RNN baseline by a significant margin. We attribute this to the temporal abstraction afforded by our skill-based learning framework. In sum, our ablation studies suggest that effective skill abstractions, coupled with mechanisms that effectively leverage the prior data, allow us to achieve strong results.

| Multi-Task Kitchen Environment | Target Tasks | |
| --- | --- | --- |
| 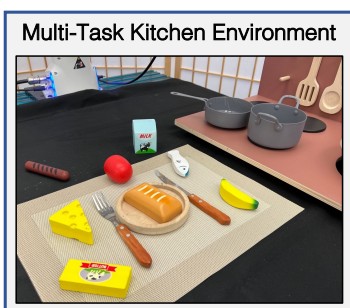 | Setting up Breakfast 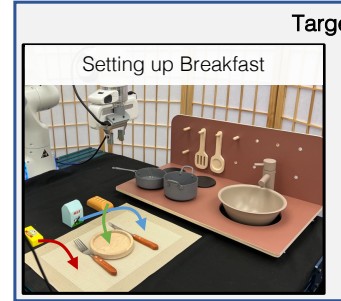 | Cooking Meal 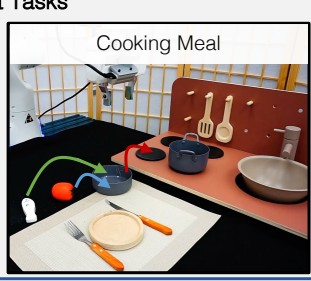 |

Figure 4: **Real World Tasks.** On the left, we illustrate the set of objects we use for collecting the play dataset. On the right shows two of three target tasks, setting up breakfast and cooking.

## 5.4 Real World Experiments

Finally, we showcase the efficacy of our method in the real world with a kitchen environment involving eight food items, receptacles, a stove, and a serving area (see Fig. 4). We first collect a play dataset of exploratory interactions involving the food items and receptacles. We consider three target tasks: (1) `Real-Breakfast`: setting up a breakfast table by placing the bread, butter, and milk in the serving area; (2) `Real-Cook`: cooking a meal by placing the fish, sausage, and tomato into the frying pan; (3) `Real-Cook-Pan`: a variant of the Real-Cook task involving placing the pan onto the stove. We collect 30 demonstrations; refer to Appendix B.3 for detailed descriptions on the tasks datasets. We evaluate SAILOR against the most competitive baseline, BC-RNN (FT) (see Appendix C.3 for our evaluation protocol). We find that while on `Real-Breakfast` both methods achieve a success rate of 76.7%, on `Real-Cook` our method significantly outperforms BC-RNN (FT) with a success rate of 73.3% vs. 23.3%, and similarly for `Real-Cook-Pan` (76.7% vs. 46.7%). To see the value of prior data we ran the No Prior ablation for the `Real-Cook` task, achieving 53.3% success rate compared to our 73.3%. Interestingly we see that the No Prior ablation largely outperforms the BC-RNN (FT) baseline on this task (23.3%) which had access to additional prior data. Overall, we observe that BC-RNN (FT) often failed to correctly grasp objects. One hypothesis for this result is that pre-training stage biases the policy to learn the multi-modal behaviors in the prior dataset, preventing the policy from learning specialized target task behaviors during the fine-tuning phase.

## 6 Limitations

While our method shows significant promise, it leaves limitations that we hope to address in future work. First, acquiring large amounts of multi-task prior data is difficult and costly. To amortize the high cost, large prior multi-task datasets should be useful in a diverse range of downstream tasks, rather than a handful. In this work, we evaluate our method in a limited set of target tasks and leave it for future work to scale up the variety of tasks. We hope (and believe) that the need for large robotic datasets will be addressed in the coming years [10, 23]. Second, our method is more computationally expensive than the BC-RNN baseline [1], due to the higher number of losses and networks used. Third, our experiments focus on domains and datasets where the prior data and target tasks are reasonably close to each other. Notably our experiments do not evaluate generalization to unseen objects between the prior and target datasets. It would be interesting to investigate methods that are tolerant to much larger domain shifts between prior and target task data.

## 7 Conclusion

We present SAILOR, a skill-based imitation learning framework for robot manipulation. Our method uses prior data to construct a latent space of predictable and consistent skill representations. It uses these latent skills as the temporal abstraction to learn policies for vision-based manipulation. Key to its effectiveness is our newly designed representation learning objectives and retrieval-based data augmentation procedure. We demonstrate that our method can solve long-horizon manipulation tasks in simulation and on physical hardware. It brings forth a data-efficient way of programming robots with new behaviors using a small number of target task demonstrations. For future work, we plan to address the limitations we discussed in the previous section and investigate the effectiveness of this approach with various forms of prior data at different scales.

**Acknowledgments**

We would like to thank Jake Grigsby, Huihan Liu, and Zhenyu Jiang for providing feedback on this manuscript. We would also like to thank Yifeng Zhu for real robot infrastructure support. We acknowledge the support of the National Science Foundation (1955523, 2145283), the Office of Naval Research (N00014-22-1-2204), and Amazon.

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

# Appendix

## A    Additional Experiments

### A.1    Ablation on prior data

We perform a more fine-grained study on the role of prior data using the CALVIN domain. We compare to variants of our method using 25% or 50% of the available prior data to see whether the quantity of prior data plays a significant role on downstream policy performance. We also compare to a variant of our method that only utilizes prior data collected from different environments than the target task environments. In the context of our CALVIN domain, the prior data spans four environments (A, B, C, D), while the target task data only spans one environment (D). Thus for this ablation we only consider data from unseen environments (A, B, C) for our prior dataset. This ablation examines the robustness of our method under environmental mismatch between the prior data and target task data. We outline all results in Table 3.

| Dataset | No prior data | 25% prior data | 50% prior data | ABC prior data | Ours |
|---|---|---|---|---|---|
| CALVIN-Setting Up | $53.3 \pm 4.0$ | $71.3 \pm 0.5$ | $76.0 \pm 2.9$ | $76.7 \pm 6.5$ | $\mathbf{77.3 \pm 3.1}$ |
| CALVIN-Cleaning Up | $60.7 \pm 3.1$ | $80.3 \pm 3.3$ | $82.3 \pm 2.1$ | $77.3 \pm 10.1$ | $\mathbf{88.0 \pm 5.1}$ |

Table 3: **Ablation on prior data.**

First, we see that increasing the size of prior data yields greater downstream policy performance. There is a significant performance gain from using no prior data to using 25% prior data, from which point increasing the amount of prior data leads to smaller gains. In addition, restricting the prior data to unseen environments still results in a meaningful performance increase, which is a promising sign that our method can operate even under controlled environmental distribution shifts between the prior and target task data.

### A.2    Ablation on retrieval metric

In this work we use $\ell_2$ distance in our latent skill space as the underlying distance measure for our retrieval operation. We also consider performing retrieval based on KL-divergence distances. ie. given two inference distributions $q_1$ and $q_2$, we compute their distance as the average forward and reverse KL divergence: $d(q_1, q_2) = \frac{1}{2}(D_{KL}(q_1||q_2) + D_{KL}(q_2||q_1))$. This metric effectively incorporates both the mean and standard deviation of the inference distributions. We compare our standard retrieval procedure with the KL-based retrieval operation in Table 4.

| Dataset | Ours | KL-based Retrieval |
|---|---|---|
| CALVIN-Setting Up | $77.3 \pm 3.1$ | $\mathbf{79.3 \pm 5.7}$ |
| CALVIN-Cleaning Up | $\mathbf{88.0 \pm 5.1}$ | $80.7 \pm 0.9$ |

Table 4: **Ablation on retrieval method.**

We do not find a significant difference between these two variants, suggesting that our method can work with alternative distance metrics for retrieval.

### A.3    Ablation on target dataset

We perform an ablation study on the size of the target dataset. We find for the CALVIN-Setting Up task that increasing the number of target task demonstrations from 30 to 100 yields an increase in success rate from $77.3\%$ to $93.3\%$ (see Table 5). Note that while it is promising that increasing the number of target task demonstrations yields an increase in success rate, this comes at the expense of additional burden for human collecting demonstrations for the target task.

| Dataset | 30 demos (Ours) | 100 demos |
|---|---|---|
| CALVIN-Setting Up | $77.3 \pm 3.1$ | $\mathbf{93.3 \pm 4.5}$ |

Table 5: **Ablation on target dataset.**

## A.4  Ablation on retrieval dataset

We perform a detailed ablation study on the quantity and quality of the retrieved data for policy learning. Recall that our retrieval procedure: (1) we first randomly sample N sub-trajectories from the prior dataset as possible retrieval candidates, (2) sort them according to their relevance to the target task, and (3) select the top r% of candidates for retrieval. We study the following ablations, which use our standard settings of N=250,000 and r=10% *unless otherwise stated*:

**No Retrieval**: N=0. Ie. we retrieve no data

**All Retrieval**: N=sizeof(prior dataset), r=100%. Ie. we retrieve the entire prior dataset. For CALVIN this is 2.3M sub-trajectories

**Random Retrieval**: instead of sorting the retrieval candidates according to relevance, we randomly select 10% of the candidates. This is a test of data quality, to see whether the relevance of the retrieved sub-trajectories to the target task matters.

**2 / 50 / 90 % Retrieval**: we retrieve r=2%, 50%, or 90% of the N candidates. This is to test whether our setting of r=10% is a good threshold for retrieval

**Large Retrieval**: N=sizeof(prior dataset). This ablation uses the same threshold r=10% as our method to perform retrieval but considers all prior sub-trajectories as retrieval candidates and thus retrieves a significantly larger quantity of data.

Note that Ours, No Retrieval, and All Retrieval are from the original submission and we include these results again for reference. We present results on the CALVIN Setting Up and Cleaning

| Dataset | Ours | No Retrieval | All Retrieval | Random Retrieval | 2% Retrieval | 50% Retrieval | 90% Retrieval | Large Retrieval |
|---|---|---|---|---|---|---|---|---|
| CALVIN-Setting Up | $77.3 \pm 3.1$ | $65.0 \pm 2.2$ | $64.3 \pm 10.8$ | $\mathbf{82.0 \pm 3.7}$ | $75.7 \pm 5.4$ | $74.3 \pm 9.9$ | $80.7 \pm 6.5$ | $79.7 \pm 0.9$ |
| CALVIN-Cleaning Up | $\mathbf{88.0 \pm 5.1}$ | $70.0 \pm 0.8$ | $65.3 \pm 7.3$ | $55.3 \pm 7.3$ | $80.7 \pm 2.9$ | $75.7 \pm 5.3$ | $46.3 \pm 20.0$ | $83.3 \pm 4.6$ |

Table 6: **Ablation on retrieval dataset.**

Up tasks in Table 6. We make the following observations for CALVIN-Cleaning Up:

- Data quality is important. The Random Retrieval retrieves the same quantity but lower quality of data as Ours. The performance significantly degrades as a result. We see the same trend from the 50 / 90% Retrieval experiments. Ie. as we increase the threshold for retrieval from r=10% to 50% and 90% (and thus decrease the quality of data) we see a consistent and significant drop in performance.

- Our standard setting of r=10% is optimal, striking the right balance between diversity and quality of data. Lower and higher thresholds (2%, 50%, 90%) perform worse.

- Retrieving larger amounts of data does not have a major impact on performance. Large Retrieval achieves performance within the margin of error as Ours.

CALVIN-Setting Up however offers a different analysis. For this task data quality does not appear to matter, as the Random retrieval, 2% / 50%, 90% Retrieval ablations all perform similarly to Ours within the margin of error. One possible explanation for this observation is that the Setting Up task involves a more diverse range of behaviors than Cleaning Up – the Setting Up task involves manipulating all components of the environment whereas the Cleaning Up task involves a subset. Another potential hypothesis is that the prior data is more biased towards behaviors seen in the Setting Up task. Because many of the behaviors in the Cleaning Up task are mirror behaviors of the Setting Up task, this may result in an unfavorable bias for the Cleaning Up task, necessitating a

retrieval procedure to filter out irrelevant behaviors.

The implication of all of these results is that the importance of retrieval may be task and dataset dependent, with some tasks being especially sensitive to the choice of retrieved data.

# B  Tasks and Datasets

## B.1  Franka Kitchen

The Franka Kitchen domain consists of a simulated 9-DoF Franka robot operating in a kitchen environment comprising a microwave, kettle, light switch, stove knobs, and a sliding and hinge cabinet. In our experiments the agent operates the robot via joint torque control resulting in a 9-dimensional action space. For observations, the agent has access to proprioceptive information consisting of the 9-dimensional joint values of the robot, in addition to RGB images from a third-person view camera and an eye-in-hand camera.

**Prior Data.** This environment is accompanied by approximately 600 human demonstrations each performing a subset of four out of seven possible subtasks: opening the microwave, turning on the light switch, turning on the top burner, turning on the bottom burner, moving the kettle, opening the hinge cabinet, and opening the sliding cabinet. We consider two prior datasets $\mathcal{D}_{\text{prior}}$: (1) using all demonstrations except the ones corresponding to the target task (`Kitchen-All`); and (2) using all demonstrations except those that involve interacting with the microwave (`Kitchen-No Microwave`). These prior datasets have 584 and 235 demonstrations, respectively.

**Target Task.** We consider one target task demonstrating a specific permutation of subtasks: opening the microwave, followed by moving the kettle, flipping on the light switch, and opening the sliding cabinet. We define task success as whether the agent has completed all of these subtasks (in no particular order). For the target dataset $\mathcal{D}_{\text{target}}$ we obtain all demonstrations in the original dataset that perform this specific permutation of subtasks, resulting in 18 demonstrations. Note that this dataset is equivalent to the `kitchen-complete-v0` dataset in the d4rl benchmark [46]. These demonstrations have an average length of 194 timesteps.

## B.2  CALVIN

The CALVIN domain consists of a simulated 7-DoF Franka robot operating in a playroom environment comprising a drawer, cubbies, two lights, and three blocks. The environment comes in four variants (see Figure 3), each with different textures, block sizes, and fixture locations. In our experiments the agent operates the robot via inverse kinematics control resulting in a 7-dimensional action space. For observations, the agent has access to proprioceptive information consisting of the robot end effector pose and gripper state, in addition to RGB images from a third-person view camera and an eye-in-hand camera.

**Prior Data.** This environment is accompanied by a large dataset of task-agnostic "play" data across all four environment variants and comprises 2.3M transitions. The play data encompass diverse behaviors, such as opening and closing drawers, turning on and off the lights, and picking, placing, and pushing blocks. We use all play data as $\mathcal{D}_{\text{prior}}$ to solve two target tasks.

**Target Tasks.** We consider two target tasks:

`CALVIN-Setting Up`: the robot must turn on the lights, retrieve the pink block from the drawer, place it on the table, and retrieve the red and blue blocks from the cubby area and place them on the table. We define task success as whether the agent has completed all of these subtasks (in no particular order). At environment resets the lights are always off, the pink block is randomly initialized inside the (closed) drawer, and the red and blue blocks are randomly initialized inside the cubby area with one block in the left region of the cubby and the other block in the right region of the cubby. For this task we collect 30 demonstrations, amounting to about half an hour of data collection. In these demonstrations, we first turn on the lights, then retrieve the pink block, then retrieve the first unoccluded block from the cubby area, then move the slider to retrieve the other block from the other side of the cubby area. These demonstrations have an average length of 584 timesteps.

`CALVIN-Cleaning Up`: the robot must open the drawer, place all three blocks into the drawer, close the drawer, and turn off the lights. We define task success as whether the agent has completed all of these subtasks (in no particular order). At environment resets the lights are always on, the drawer is closed, and the three blocks are randomly placed in left, center, and right regions of the table. For this task we collect 30 demonstrations, amounting to about half an hour of data collection. In these demonstrations, we first open the drawer, then place the blocks on by one into the drawer from right

to left, then close the drawer, and finally turn off the lights. These demonstrations have an average length of 572 timesteps.

## B.3 Real World Kitchen

We designed a real world kitchen environment to study the utility of our method on physical hardware. Our kitchen environment comprises a Flexa toy kitchen[2], a set of toy food items[3], a number of serving items (placemat, plate, knife, fork), and a small pot and pan that we purchased from a local store. We use a 7-DoF Franka Emika Panda robot which is operated via Operational Space Control (OSC) [49]. We found OSC to be a fitting choice, as it offers task-space compliant behavior that makes for a more intuitive data collection experience. We restrict the OSC controller to the position and yaw of the end effector[4], which combined with the gripper controller results in a 5-dimensional action space. For observations, the agent has access to proprioceptive information consisting of the robot end effector pose and gripper state, in addition to RGB images from a third-person view camera and an eye-in-hand camera.

**Prior Data.** We collect a large prior dataset of task-agnostic play behaviors involving the food items and the pot and pan. Overall our prior dataset involves 150 trajectories each with approximately 2,000 timesteps, resulting in approximately 300,000 total timesteps. For each trajectory we first initialize the scene by randomly sampling four out of eight food items (milk, bread, butter, sausage, fish, tomato, banana, cheese) and randomly placing these four items around the serving area. We also randomly initialize the pot and pan on the two front stove burners or occasionally place one on the table next to the serving area. We then randomly pick and place food items either on the table, the serving area, or the pot and pan. We also occasionally pick and place the pot or pan to the table or stove burners.

**Target Tasks.** We consider three target tasks:

`Real-Breakfast`: the objective of this task is to place the bread, butter, and milk from the table onto the serving area. These food items are initialized randomly in the vicinity of three possible locations on the table: the left, center, and right of the region preceding the serving area. We consider two possible permutations for the placement of object onto these three regions (in left-center-right format): butter-bread-milk, bread-butter-milk, and butter-milk-bread. The pots and pans are initialized on the front stove burners. We define task success as whether the robot has (in no particular order) placed the bread onto the plate, the butter to the left of the plate on the placemat, and the milk to the right of the plate on the placemat. For this task we collect 30 demonstrations, amounting to about half an hour of data collection. In these demonstrations we place the bread, butter, and milk in order onto their corresponding goal locations. These demonstrations have an average length of 546 timesteps.

`Real-Cook`: the objective of this task is to place the fish, sausage, and tomato from the table into the pan. These food items are initialized randomly in the vicinity of three possible locations on the table: the left, center, and right of the region preceding the serving area. We consider three possible permutations for the placement of object onto these three regions (in left-center-right format): fish-sausage-tomato, sausage-fish-tomato, and fish-tomato-sausage. The pots and pans are initialized on the front stove burners. We define task success as whether the robot has (in no particular order) placed these three items into the pan. For this task we collect 30 demonstrations, amounting to about half an hour of data collection. In these demonstrations we place the food items from left to right (in order) into the pan. These demonstrations have an average length of 548 timesteps.

`Real-Setup-Pan`: the objective of this task is to place the pan from the table onto the stove and subsequently place the fish and sausage into the pan. The pan is initialized randomly in the vicinity of the right region of the table preceding the serving area. The food items are initialized randomly in the vicinity of the left and center regions of the table preceding the serving area. We consider three possible permutations for the placement of the objects onto these three regions (in left-center-right format): fish-tomato-pan and tomato-fish-pan. The pot is initialized on the front stove burners. We define task success as whether the robot has (in no particular order) placed the pan onto the stove

---

[2] https://flexa-usa.com/collections/play/products/toys-the-kitchen
[3] https://www.amazon.com/Melissa-Doug-Food-Groups-Hand-Painted/dp/B0000BX8MA
[4] We did not find the roll and pitch actuation to be necessary for our real world tasks and we opted for a simpler action space.

and the two food items into the pan. For this task we collect 30 demonstrations, amounting to about half an hour of data collection.

# C Implementation Details

## C.1 Model Architecture

We elaborate further on our method architecture outlined in Figure 2. Our implementation is based on top of robomimic[5], a recent open source codebase with extensive benchmarking results across a number of imitation learning algorithms. We adopted the same neural modules (same RNN backbone, VAE, visual perception encoders) for our algorithm, and in fact our BC-RNN baseline uses the exact implementation from robomimic.

Our model specifically consists of five neural network modules: four networks for the skill model comprising an RNN encoder, an RNN decoder, a feedforward VAE prior[6], and a feedforward temporal prediction network; and one RNN network for the policy.

**Observation Encoder.** Four out of the five modules described above take observation inputs (among other potential inputs), and each of these modules is equipped with an observation encoder to process these observations. The observation encoder specifically consists of ResNet-18 backbones [50] to encode the third-person image and eye-in-hand image, and a multi-layer perceptron (MLP) for all remaining low-dimensional observational inputs. Note that we pre-process the ResNet inputs with random cropping and post-process the outputs with a Spatial Softmax [51] pooling layer. After processing the image and low-dimensional observation inputs we concatenate the resulting outputs to form one unified observation encoding. Note that our RNN encoder, RNN decoder, and RNN policy process a sequence observations individually using the observation encoder and then processes these encoded observations into one unified representation with a recurrent neural network.

**Skill Model.** The skill model is a Variational Autoencoder that encodes sub-trajectories into a latent skill representation and decodes information back into the actions of sub-trajectories. The skill encoder and decoder are RNNs with a 2-layer LSTM followed by a 2-layer MLP, while the VAE prior and temporal prediction network are 2-layer MLPs.

**Policy.** The policy is a 2-layer LSTM network that maps a history of $F$ observations into a latent skill $z$. We also condition the policy on a dataset id $\in \{0, 1\}$ to indicate whether the policy is optimized on the target dataset or the retrieval dataset, to prevent potential interference between the target and retrieved data (see Algorithm 2 for additional details). Note that we can extend our policy to incorporate fine-grained goal information by conditioning on additional context information such as goal images or language goals [44, 52, 23].

## C.2 Training

Our algorithm consists of two phases. In the first phase we pre-train our skill model on sub-trajectories in $\mathcal{D}_{\text{prior}}$ (see Algorithm 1 for further details[7]). In the subsequent phase we are given the target dataset $\mathcal{D}_{\text{target}}$ and we proceed to learning the policy and fine-tuning the skill model. Before we perform policy learning, we first retrieve sub-trajectories in $\mathcal{D}_{\text{prior}}$ that have similar embeddings to those in $\mathcal{D}_{\text{target}}$. We aggregate these retrieved embeddings into our retrieval dataset $\mathcal{D}_{\text{ret}}$. We then proceed to train the policy jointly on embeddings from $\mathcal{D}_{\text{target}}$ and $\mathcal{D}_{\text{ret}}$. At the same time we continue to fine-tune the skill model with sub-trajectories sampled from both $\mathcal{D}_{\text{prior}}$ and $\mathcal{D}_{\text{target}}$. We summarize these steps in Algorithm 2.

We sample fixed-length sub-trajectories uniformly at random to train our model, following recent skill-based imitation learning works [6, 9, 7]. More specifically, for each dataset we concatenate all trajectories into one continuous stream of data and uniformly sample sub-trajectories from this stream. Note that this can result in sampling overlapping sub-trajectories. For training the policy we additionally train on the frame stack of observations preceding the sampled sub-trajectory. There are some edge cases, such as when the sub-trajectory intersects with the next trajectory and when

---

[5]https://github.com/ARISE-Initiative/robomimic

[6]we also utilize a feedforward deterministic inverse dynamics model but we found that it does not lead to a significant change in downstream policy learning results

[7]in our code we also have a slowness term to ensure that two nearby sub-trajectories have similar skill embeddings. We did not find this feature to have a noticeable impact on downstream policy performance and therefore we omit it from the algorithm pseudocode for simplicity.

the frame-stack intersects with the previous trajectory. We deal with these cases by padding all data from the offending consecutive trajectory with the first / last observation of the current trajectory.

---

**Algorithm 1** Skill pretraining

---

**Input:** prior dataset $\mathcal{D}_{\text{prior}}$

1: **while not** done **do**
2:     Sample $\tau = \{(o_t, a_t)\}_{t=0}^{H-1} \sim \mathcal{D}_{\text{prior}}$
3:     Sample temporal offset $d \in [-50, 50]$ and obtain $\tau'$ shifted $d$ timesteps from $\tau$
4:     $\mu_\tau, \Sigma_\tau \leftarrow q_\phi(\tau)$                               // Encode $\tau$
5:     $\mu_{\tau'}, \Sigma_{\tau'} \leftarrow q_\phi(\tau')$                           // Encode $\tau'$
6:     $z \sim \mathcal{N}(\mu_\tau, \Sigma_\tau)$                            // Sample latent z
7:     $\hat{a}_t \leftarrow p_\psi(o_t, z)$                          // Decode actions through RNN
8:     $\hat{d} \leftarrow m_\omega(\mu_\tau, \mu_{\tau'})$                    // Predict temporal distance
9:     $\mathcal{L}_{\text{VAE}} \leftarrow \|\hat{a} - a\|^2 + \beta \cdot D_{KL}(q_\phi(\tau)\|p_\theta(o_0, o_H))$   // Compute VAE Loss
10:    $\mathcal{L}_{TC} \leftarrow \alpha \cdot (\hat{d} - d)^2$                   // Compute TC Loss
11:    $\mathcal{L}_{\text{Skill}} \leftarrow \mathcal{L}_{\text{VAE}} + \mathcal{L}_{\text{TC}}$
12:    update $\phi, \psi, \theta, \omega$ on $\mathcal{L}_{\text{Skill}}$ via gradient descent
13: **end while**

---

---

**Algorithm 2** Policy learning and skill fine-tuning

---

**Input:** prior dataset $\mathcal{D}_{\text{prior}}$, target dataset $\mathcal{D}_{\text{target}}$, pre-trained skill encoder $q_\phi$

1:  // Obtain retrieval dataset
2: $Z_{\text{prior}} \leftarrow \{\mu(q_\phi(\tau_i))\}_{i=1}^N, \tau_i \sim \mathcal{D}_{\text{prior}}$       // Sample and encode N sub-trajs from $\mathcal{D}_{\text{prior}}$
3: $Z_{\text{target}} \leftarrow \{\mu(q_\phi(\tau_j))\}_{j=1}^M, \tau_j \sim \mathcal{D}_{\text{target}}$     // Sample and encode M sub-trajs from $\mathcal{D}_{\text{target}}$
4: $D_{ij} = \|Z_{\text{prior}}^i - Z_{\text{prior}}^j\|_2$             // Compute all pairwise encoding distances
5: $D\_min_i = \min_j(D_{ij})$  // Find closest target sub-traj for each prior sub-traj
6: $K \leftarrow \texttt{argsort(D\_min)[:n]}$  // Compute list of indices in $Z_{\text{target}}$ with minimal distance
7: $\mathcal{D}_{\text{ret}} \leftarrow \{(o_{fs}^k, Z_{\text{prior}}^k)\}_{k \in K}$  // Retrieve skill embeddings and their preceding observations
8:
9:  // Train policy
10: **while not** done **do**
11:    Sample $(o_{fs}, z) \sim \mathcal{D}_{\text{target}}$               // target dataset sub-traj encoding and frame stack
12:    Sample $(o'_{fs}, z') \sim \mathcal{D}_{\text{ret}}$             // retrieval dataset sub-traj encoding and frame stack
13:    $\hat{z} \leftarrow \pi(o_{fs}, \text{id} = 0)$               // predict skill for target dataset sub-traj
14:    $\hat{z}' \leftarrow \pi(o'_{fs}, \text{id} = 1)$             // predict skill for retrieval dataset sub-traj
15:    $\mathcal{L}_{\text{Policy}} \leftarrow (\hat{z} - z)^2 + \gamma \cdot (\hat{z}' - z')^2$     // Compute Policy Loss
16:    update $\pi$ on $\mathcal{L}_{\text{Policy}}$ via gradient descent
17:    fine-tune $\mathcal{L}_{\text{Skill}}$ on sub-trajectories sampled from $\mathcal{D}_{\text{prior}}$ and $\mathcal{D}_{\text{target}}$  // see Algorithm 1
18: **end while**

---

### C.3 Evaluation

To perform a policy rollout, we first obtain a skill $z = \pi(o'_{fs}, \text{id} = 0)$ from the policy. We execute this skill with our closed-loop skill decoder $p_\psi(o_t, z)$ for $H$ timesteps and we subsequently repeat the process by obtaining a new skill from the policy. Note that we do not preempt skill execution; we execute all $H$ timesetps until completion[8]. We terminate the episode either when the agent has successfully solved the task or if the agent has exceeded the time budget for the rollout. We assess each episode based on whether the agent successfully solved the task in the allotted time budget. While other metric also exist (time to complete task), we chose binary success for its popularity and relative simplicity. We elaborate further on our evaluation protocol:

---

[8] We believe this is reasonable choice, as (1) the closed-loop skill decoder can react to current environment conditions during skill execution, and (2) the policy is still operating at high frequency and can react accordingly

**Simulation Experiments**: We evaluate the success rate across 3 seeds (unless otherwise noted) and report the average and standard deviation across all seeds. To evaluate a seed, we perform 100 policy rollouts every $n$ checkpoints and record the success rate for each checkpoint. We then record the success rate for the seed as the highest success rate across all checkpoints evaluated for that seed[9].

**Real World Experiments**: Due to the challenges of real-world evaluation we only evaluate 1 seed for each baseline. To evaluate an experiment, we perform an initial evaluation of different policy checkpoints, evaluating each checkpoint for only a few trials. Upon choosing the most promising checkpoint we perform 30 rollouts and report the success rate over these rollouts.

## C.4 Baselines

All baselines are implemented in the robomimic codebase for fair comparison. We briefly elaborate on these implementations as follows:

**BC-RNN**: We use the default implementation of BC-RNN in robomimic and we use identical hyperparameters as those reported in the robomimic study paper [1].

**BC-RNN (FT)**: We use an identical architecture and identical hyperparameters as BC-RNN. We first train the baseline on $\mathcal{D}_{\text{prior}}$ and subsequently fine-tune on $\mathcal{D}_{\text{target}}$ via a second stage of training. We also experimented with jointly training a task-conditioned BC-RNN policy in $\mathcal{D}_{\text{prior}}$ and $\mathcal{D}_{\text{target}}$ but we found that it yielded very poor performance due to the multi-modality of actions in the prior data.

**BC-RNN (R3M)**: We use an identical architecture and identical hyperparameters as BC-RNN but with a pretrained R3M visual representation. We specifically replace the weights of our ResNet-18 networks with the pretrained ResNet-18 weights from R3M[10]. We follow the same practice from the R3M paper and we freeze the pretrained ResNet weights during downstream imitation learning.

**IQL**: We base our implementation off of the publicly available PyTorch implementation of IQL[11].

**IQL (UDS)**: We make small modification to our IQL implementation. For each batch that we sample from $\mathcal{D}_{\text{target}}$, we also sample an equivalent-size batch from $\mathcal{D}_{\text{prior}}$ with the rewards set to $0$. We then perform gradient updates on the aggregated data from both of these batches.

**FIST**: We use the same underlying skill model as our method but a semi-parametric policy in place of our parametric neural network policy. We use an identical scheme for the semi-parametric policy as the FIST paper [7].

## C.5 Environment Implementation

### C.5.1 Gripper Logic

We elaborate on the gripper logic in our environments. The gripper state is either the position of the gripper fingers (Franka Kitchen, Real World Kitchen) or the opening width of the gripper (CALVIN). The gripper action is a continuous 1-D variable, and we interpret this as either opening (if $< 0$) or closing (if $> 0$). The gripper is controlled via position control. When the agent specifies a closing action the position target of the controller is set to close the gripper fingers all the way (and for opening the target is set to open the gripper fingers all the way). There are limits on the force and velocity of the fingers in order to ensure gripper stability.

---

[9]this is the same evaluation protocol used in [1]

[10]https://github.com/facebookresearch/r3m

[11]https://github.com/rail-berkeley/rlkit/tree/master/examples/iql

# D   Hyperparameters

We adopted a similar set of hyperparameters as the BC-RNN baseline from robomimic—we used the same LSTM settings, batch size, and RNN policy history length of $F = 10$. We did experiment with different choices of sub-trajectory lengths for the skill model ($H = \{1, 5, 10, 25\}$) and found that $H = 10$ performs optimally. Longer horizons may be helpful in some settings, however we hypothesize that RNN-based architectures lack the capacity to accurately predict actions over significantly longer horizons. It would be interesting to investigate if the optimal $H$ changes under a Transformer-based [53] architecture.

| Hyper-parameter | Value |
|---|---|
| Skill encoder: # LSTM hidden units | 1000 |
| Skill encoder: MLP hidden sizes | $1024, 1024$ |
| Skill decoder: # LSTM hidden units | 1000 |
| Skill decoder: MLP hidden sizes | $1024, 1024$ |
| Skill prior: hidden sizes | $1024, 1024$ |
| TC: hidden sizes | $128, 128$ |
| Policy: # LSTM hidden units | 1000 |
| Sub-trajectory length $H$ | 10 |
| Skill latent dimension | 64 |
| Skill KL weight $\beta$ | $1e{-}5$ |
| TC weight $\alpha$ | $1e{-}6$ |
| Retrieval weight $\gamma$ | 0.15 for Real tasks, else 1.0 |
| # observation history frames $F$ | 10 |
| Batch size | 16 |
| Optimizer | Adam [54] |
| Learning rate: skill VAE | $5e{-}4$ |
| Learning rate: TP | $1e{-}4$ |
| Learning rate: Policy | $1e{-}3$ |
| Retrieval: max # $\mathcal{D}_{\text{prior}}$ samples $N$ | 300,000 for Real tasks, else 250,000 |
| Retrieval: max # $\mathcal{D}_{\text{target}}$ samples $M$ | 2,500 |
| % of samples chosen for retrieval | 5% for Real tasks, else 10% |

Table 7: Hyperparameters for our method.

| Method | # Epochs | Evaluation checkpoints freq $n$ |
|---|---|---|
| BC-RNN | 800 | 40 |
| BC-RNN (FT): phase 1 | Franka Kitchen: 300 
 CALVIN: 600 
 Real tasks: 300 | – |
| BC-RNN (FT): phase 2 | Franka Kitchen: 400 
 CALVIN: 400 
 Real tasks: 600 | 20 
 20 
 50 |
| BC-RNN (R3M) | 800 | 40 |
| IQL | 800 | 40 |
| IQL (UDS) | 800 | 40 |
| FIST: phase 1 | Franka Kitchen: 300 
 CALVIN: 600 | – |
| FIST: phase 2 | 200 | 10 |
| Ours: phase 1 | Franka Kitchen: 300 
 CALVIN: 600 
 Real tasks: 200 | – |
| Ours: phase 2 | 200 | 10 |

Table 8: Training and Evaluation Hyperparameters.

| Task | Image Size | Rollout Length |
|------|:---:|:---:|
| Franka Kitchen | $84 \times 84$ | 280 |
| CALVIN: Setting Up | $84 \times 84$ | 1000 |
| CALVIN: Cleaning Up | $84 \times 84$ | 1000 |
| Real Breakfast | $128 \times 128$ | 1500 |
| Real Cook | $128 \times 128$ | 1500 |

Table 9: Task Hyperparameters.

