# OpenReview forum: "Learning and Retrieval from Prior Data for Skill-based Imitation Learning"
_robot-learning.org/CoRL/2022/Conference — CoRL 2022 Poster_

### Official Review · Reviewer_XCRj · 2022-07-20

**Originality:** Very Good
**Technical Quality:** Very Good
**Clarity Of Presentation:** Very Good
**Impact:** 3

**Recommendation:**

Weak Accept: I recommend accepting the paper, but will not argue for my recommendation if the majority of other reviewers have a different opinion.

**Summary:**

This paper presented Skill-Augmented Imitation Learning with prior Retrieval
(SAILOR), for improving task learning using skill learning from prior data and a small amount of target task data. The two key additions of this paper are: latent skill learning with an auxiliary task that encourages skills that are temporally similar to be close in latent skill space, and data retrieval that compares (in the latent space) the skills of the prior data to the target task dataset and limits the sampling of skills during downstream policy learning to those close to demonstrations on the target task.


**Issues:**

No figures are referenced in the text. It is not known when the figures back up what is written, which breaks up readability.

The symbols are confusing in parts.
- “p” is used as transition probably, initial state distribution, skill decoder, and learned prior.
- Figure 2 could be clearer. Is the “temporal consistency” block the same as “m_ω”? Is “phi” the skill encoder reported as “q_phi” in the paper? Is “psi” the same as “p_psi”?
- It might be worth changing the font in figure 2 to match what is in the text to improve first-glance readability.

The reference list often shows the arxiv URL, but not the publication venue if the paper was accepted. For example: “LASER: Learning a Latent Action Space for Efficient Reinforcement Learning” was accepted at ICRA 2021. At first glance the reader may think many of the references are not from peer-reviewed sources.

Spelling / Grammar / Typos:

“to a specific types of tasks”

“These includes task-conditioned”

“our goal is learn a policy”

“representations of behavior is critical”

“train the policy the aggregated set”

“prior dataset that similar”

Very minor, (to improve readability):
“In contrast to other skill-based imitation learning (see
Sec. 3) approaches,” -> “In contrast to other skill-based imitation learning approaches (see
Sec. 3),”

“Yet despite their appeal, after experimenting with one such approach, R3M [28], we found that it can sometimes hinder downstream performance.”  -> as this was encountered later in this paper, maybe change the wording to “for the tasks explored in this paper we found that it can sometimes hinder downstream performance.”


**Quality Of The Limitations Section:**

Limitations are addressed clearly

**Reviewer Expertise:**

4: The reviewer is confident but not absolutely certain that the evaluation is correct

**Robotics Focus:**

Sufficient demonstration on hardware

**Strengths And Weaknesses:**

This paper is well presented, clear, and easy to follow. The contributions are minor practical enhancements of existing approaches, however, the tasks are well selected to show a significant improvement over existing methods. The comparisons seem relevant, and an ablation study shows the effect of each component of the method. Finally, the real-world experiments round off this paper nicely.

I think the main thing that is missing is a deeper evaluation of the two contributions. For example, it would be nice to show that regions in the latent space are indeed close in temporal space, and that behaviours retrieved are indeed close to behaviours from the target task dataset. The ablation study does enough to show these contributions are effective, though it would be a nice addition to see the effect of these methods practically, and also beneficial to the community that may adopt these approaches. This could be as simple as showing samples in each case and discussing the edge cases.


**Summary Of Recommendation:**

As mentioned, a deeper analysis of the effectiveness of each of the contributions would enhance the paper.

---

> ### Author Response · Authors · 2022-08-26
> **Author Response to Reviewer XCRj**
>
> Dear Reviewer,
>
> Thank you very much for your positive and constructive review! We are happy to hear that you found our paper clear to understand, that you found our comparisons and ablations comprehensive, and that you appreciate our real world evaluations. Following your suggestions we have made a number of changes to our paper to include additional analysis on our contributions and to improve the presentation of our paper. We address your concerns below.
>
> > This could be as simple as showing samples in each case and discussing the edge cases.
>
> Indeed, we agree that this is a great idea. In the newly attached supplementary materials we present (1) the original demonstrations for the CALVIN Setting Up and Cleaning Up tasks (see videos/target_task_demos folder), (2) examples of retrieved sub-trajectories for each task (see videos/retrieval_ours folder), and (3) random sub-trajectories from the prior dataset for reference (see videos/retrieval_random folder). Qualitatively we see that while the random sub-trajectories demonstrate a wide range of behaviors, the retrieved sub-trajectories demonstrate more focused behaviors that are relevant to each target task. For example, for Setting Up we frequently see the robot turning on lights, opening the cubby, and approaching or placing objects. For the Cleaning Up task we see the agent turning off lights, placing objects into the drawer, and closing the drawer. At the same time we also see cases of irrelevant behaviors, such as the agent turning on the LED for the Cleaning Up task. The implication of these visualizations is that the skill embeddings are an approximate heuristic for skill similarity.
>
> > it would be nice to show that regions in the latent space are indeed close in temporal space, and that behaviours retrieved are indeed close to behaviours from the target task dataset
>
> Thank you for the suggestion! Following the visualizations we reference above, we have also added visualizations of the retrieved sub-trajectories using a skill model without the temporal consistency term (see the retrieval_no_tc folder). Based on our subjective interpretation, the retrieved segments appear similar with and without temporal consistency. We will leave these videos for reviewers to analyze and arrive at their own qualitative interpretations.
>
> > No figures are referenced in the text. It is not known when the figures back up what is written, which breaks up readability.
>
> Thank you for pointing this issue out. We have added references to figures and appendix sections throughout the paper to improve readability.
>
> > “p” is used as transition probably, initial state distribution, skill decoder, and learned prior.
>
> Thank you for the feedback. Whenever possible we attempted to use the same symbol conventions as prior literature to avoid potential confusion. “p” is unfortunately the standard symbol for all of the components that you mentioned above, as it denotes probability. We hope that readers will correctly identify the symbols based on the symbol subscripts.
>
> > Figure 2 could be clearer. Is the “temporal consistency” block the same as “m_ω”? Is “phi” the skill encoder reported as “q_phi” in the paper? Is “psi” the same as “p_psi”?
>
> Indeed there are some inconsistencies in the figure from the rest of the paper. Thank you for pointing this out! We have updated the figure to match the conventions in the rest of the paper.
>
> > It might be worth changing the font in figure 2 to match what is in the text to improve first-glance readability.
>
> Thank you for the suggestion. We were also hoping to match the fronts but unfortunately were unable to locate the same font in our software editing tool for making the figures. We are hoping to resolve the issue for the final version of the paper.
>
> > The reference list often shows the arxiv URL, but not the publication venue if the paper was accepted.
>
> Thank you for pointing this issue out. We have updated all references to reflect the most recent publication venues.
>
> > Spelling / Grammar / Typos:
>
> Thank you very much for pointing out these details. In our updated manuscript we have fixed all of these issues.

---

> > ### Comment · Reviewer_XCRj · 2022-08-26
> > **Response to authors**
> >
> > Thank you for your improvements. I have no further concerns. Best of luck for the next stage.

---

> > > ### Author Response · Authors · 2022-08-28
> > > **Thank you!**
> > >
> > > We are happy to hear your prompt response, thank you!

---

### Official Review · Reviewer_Qe4M · 2022-07-26

**Originality:** Good
**Technical Quality:** Good
**Clarity Of Presentation:** Excellent
**Impact:** 3

**Recommendation:**

Weak Accept: I recommend accepting the paper, but will not argue for my recommendation if the majority of other reviewers have a different opinion.

**Summary:**

This paper addresses the problem of imitation learning. Specifically, this work assumes access to a prior dataset that contains task-agnostic play data or demonstrations solving different tasks and extracts temporally-extended skills from it. Then, a small sef of demonstrations for the target task are provided for learning the target tasks. To this end, the paper proposes a framework that is built upon existing skill-based imitation learning methods, while introducing two techniques: (1) when learning skills, additionally optimizes a temporal consistency term that encourages the learned skill space, and (2) when learning policy, leverages the prior demonstrations which are similar to the target demonstrations. The experiments on two robot manipulation domains (Franka Kitchen and CALVIN) show that the proposed framework outperforms baselines, including BC, IQL, and FIST. Ablation studies justify the effectiveness of the two proposed techniques.

**Issues:**

Described in the Strengths And Weaknesses section.

**Quality Of The Limitations Section:**

Limitations are addressed clearly

**Reviewer Expertise:**

4: The reviewer is confident but not absolutely certain that the evaluation is correct

**Robotics Focus:**

Sufficient demonstration on hardware

**Strengths And Weaknesses:**

## Paper strengths and contributions
**Motivation and intuition**
The motivation for leveraging prior datasets to accelerate target task learning is convincing.

**Technical contribution**
- Encouraging the learned skill space to capture temporal difference seems effective and this paper presents a reasonable implementation to achieve it.
- Augmenting the target dataset using similar demonstrations from the prior dataset seems simple yet helpful.

**Clarity**
The overall writing is clear. The authors utilize figures well to illustrate the ideas.

**Ablation study**
Ablation studies are comprehensive. The proposed framework incorporates two techniques and the provided ablation studies are helpful for analyzing the effectiveness of each of them.

**Experimental results**
- The presentation of the experimental results is clear.
- The experimental results on two robot manipulation domains (Franka Kitchen and CALVIN) show that the proposed framework outperforms baselines, including BC, IQL, and FIST.

**Reproducibility**
Given the clear description in the main paper and the details provided in the appendix, I believe reproducing the results is possible.

## Paper weaknesses and questions

**Navigation and locomotion domains**
This work evaluates the proposed framework in two robot manipulation domains. It would be more convincing if the experiments were also conducted in navigation or locomotion domains. I believe the proposed temporal consistency term would yield an even larger improvement in navigation domains.

**Implicitly predicting time step difference**
This work proposes to predict the time step difference between two sub-trajectories for encouraging temporal consistency. I wonder if implicitly learning this (such as performing contrastive learning where the model select which sub-trajectories are closer in terms of temporal difference, see "Time-Contrastive Networks: Self-Supervised Learning from Video" by Sermanet et al.) would also yield improvement.

**Kitchen-No Microwave vs. Kitchen-All**
I wonder why FIST and IQL (UDS) perform better on Kitchen-No Microwave, which is supposed to be harder compared to Kitchen-All.

**Similarity between the prior dataset and the target dataset**
This work assumes that the prior dataset and the target dataset are similar enough so that learning from some of the prior demonstrations is helpful. I found this assumption a little too strong.

**Similarity between trajectories**
This work proposes to use $L_2$ in the learned skill space to measure the similarity between two trajectories. At first glance, it does not look like a good idea. Yet, the improved performance backs it up. I feel I would get more convinced if prior demonstrations with high and low similarity are qualitatively presented.

**$\mu$ in eq(2)**
I was a bit confused reading Eq. (2) since $\mu$ in Eq. (2) is not introduced when it is first mentioned there. It is only mentioned in L157.

====== After rebuttal ======

I thank the authors for the response and the revision, which addresses many of my concerns. I have carefully read other reviews and decided to keep my original rating (weak accept).

**Summary Of Recommendation:**

I believe this work studies an important and active research direction (i.e. skill-based learning in RL) and proposes two interesting reasonable techniques. The experimental results justify the effectiveness of the proposed techniques. Yet, I have a few concerns (stated above) such as the limited scope of the evaluation (only in robot manipulation), I am leaning toward accepting this paper.

---

> ### Author Response · Authors · 2022-08-26
> **Author Response to Reviewer Qe4M (1/2)**
>
> Dear Reviewer,
>
> Thank you very much for your thorough and constructive review! We appreciate your positive comments regarding the motivation and clarity of our work, and we appreciate that you found our ablation studies to be comprehensive. Please note that we have added a number of new experiments and visualizations and have updated the manuscript. We address your concerns as follows.
>
> > It would be more convincing if the experiments were also conducted in navigation or locomotion domains.
>
> Thank you for the suggestion! Due to time constraints in the rebuttal phase we were unable to run experiments on additional navigation and locomotion domains. However, we highlight that our method can be applied across a wide array of applications spanning manipulation, navigation, and even decision-making in domains outside of robotics. We have designed our codebase such that researchers from diverse areas of robotics can easily interface their tasks and datasets with our algorithm. We plan to open source our code in the near future.
>
> > This work proposes to predict the time step difference between two sub-trajectories for encouraging temporal consistency. I wonder if implicitly learning this (such as performing contrastive learning where the model select which sub-trajectories are closer in terms of temporal difference, see "Time-Contrastive Networks: Self-Supervised Learning from Video" by Sermanet et al.) would also yield improvement.
>
> Thank you for the comment! Indeed, shaping the latent representation via a contrastive learning objective is a promising alternative to the scheme we used in our work. One possibility would be to label temporally close pairs of sub-trajectories as positive examples while labeling temporally distant (or random) pairs as negative examples. We anticipate that this objective can yield better policies for downstream target tasks. Per your comment we have updated the manuscript with a discussion on contrastive representation learning objectives.
>
> > Kitchen-No Microwave vs. Kitchen-All I wonder why FIST and IQL (UDS) perform better on Kitchen-No Microwave, which is supposed to be harder compared to Kitchen-All.
>
> This is a great observation. We noticed that for the Franka Kitchen domain our IQL (UDS) implementation could benefit from additional tuning of hyperparameters (we had already done extensive hyperparameter tuning for UDS). The updated hyperparameters attain a similar success rate as before but much more reasonable Q value estimates. The updated results show that Kitchen-All attains higher performance than Kitchen-No-Microwave.
>
> However for FIST Kitchen-No Microwave outperforms Kitchen-All by a significantly larger margin. One possible explanation for this result is that the underlying model has greater optimization difficulties on the Kitchen-All dataset due to the greater scope and diversity of this dataset. In fact, from our experiment training logs we see that the skill losses are higher for Kitchen-All than Kitchen-No Microwave. In particular we see higher losses for the VAE inverse dynamics prior, resulting in skills that have lower fidelity in completing subgoals for Kitchen-All. Note that our method uses the skill model in a different manner as FIST (we use the skill encoder to determine which skill the policy should execute while FIST uses the VAE prior), and may be affected by this issue in different ways. An alternative hypothesis is that it is more difficult to optimize the contrastive distance model in FIST with Kitchen-All due to the increased diversity of this dataset (please refer to the FIST paper [1] for more details on this distance model). Further investigation is needed to examine these hypotheses.
>
> [1] Hierarchical few-shot imitation with skill transition models, Hakhamaneshi et al., ICLR 2022
>
> > This work assumes that the prior dataset and the target dataset are similar enough so that learning from some of the prior demonstrations is helpful. I found this assumption a little too strong.
>
> We fully understand your concern and we acknowledge in the Limitations section that this is one of the main limitations of our approach: “our experiments focus on domains and datasets where the prior data and test tasks are reasonably close to each other (same robot hardware and similar workspace). It would be interesting to investigate methods that are tolerant to much larger domain shifts between prior and test data.” Please let us know if you have additional suggestions on communicating this limitation.

---

> > ### Author Response · Authors · 2022-08-26
> > **Author Response to Reviewer Qe4M (2/2)**
> >
> > > This work proposes to use  in the learned skill space to measure the similarity between two trajectories. At first glance, it does not look like a good idea. Yet, the improved performance backs it up. I feel I would get more convinced if prior demonstrations with high and low similarity are qualitatively presented.
> >
> > Thank you for the suggestion! We agree that this is a great idea. In the newly attached supplementary materials we present (1) the original demonstrations for the CALVIN Setting Up and Cleaning Up tasks (see videos/target_task_demos folder), (2) examples of retrieved sub-trajectories for each task (see videos/retrieval_ours folder), and (3) random sub-trajectories from the prior dataset for reference (see videos/retrieval_random folder). Qualitatively we see that while the random sub-trajectories demonstrate a wide range of behaviors, the retrieved sub-trajectories demonstrate more focused behaviors that are relevant to each target task. For example, for Setting Up we frequently see the robot turning on lights, opening the cubby, and approaching or placing objects. For the Cleaning Up task we see the agent turning off lights, placing objects into the drawer, and closing the drawer. At the same time we also see cases of irrelevant behaviors, such as the agent turning on the LED for the Cleaning Up task. The implication of these visualizations is that the skill embeddings are an approximate heuristic for skill similarity.
> >
> > > \mu in eq(2) I was a bit confused reading Eq. (2) since \mu in Eq. (2) is not introduced when it is first mentioned there. It is only mentioned in L157.
> >
> > Thank you for raising this issue. We have updated the manuscript to resolve this ambiguity.
> >
> > > Related work While the related work in its current form is sufficient, I believe discussing the following would make it even more comprehensive.
> >
> > Thank you for the feedback. We have updated our related works section with a detailed discussion on skill-based imitation learning. We discuss works that segment demonstrations into variable-length segments using variational autoencoders, Bayesian online changepoint detection, and sequence alignment, among other approaches. We additionally discuss how our work fits within the context of these prior approaches.

---

### Official Review · Reviewer_m8W5 · 2022-07-29

**Originality:** Very Good
**Technical Quality:** Very Good
**Clarity Of Presentation:** Good
**Impact:** 4

**Recommendation:**

Weak Accept: I recommend accepting the paper, but will not argue for my recommendation if the majority of other reviewers have a different opinion.

**Summary:**

The work introduces a new method for skill-based imitation learning which can exploit the availability of prior data. It proposes a two-step method. In the first step a "latent skill spaced" is learned using a variational objective augmented with a temporal consistency term which enforces the predictability of the temporal distance of sub-trajectories. Second, a policy is learned by behavioral cloning on skill level using both data from the target task at hand as well as a subset of the prior data which is built based on the distances of embedded skills.
The resulting method is evaluated on two simulated and one real-world experiment and ablations are performed on one of the settings.

**Issues:**

I believe the paper would heavily benefit from more rigor and details in the method section. To be clear, I don't think any of this is "wrong" or unreasonable but figuring that out and piecing it together should not be left to the reader but be more clearly stated:

- VAE loss objective (Equation 1): What exactly is lower bounded ($\sum_{t=0}^{H-1} log p(a_t|o_t)$, what are the independence assumptions/graphical models underlying the generative and inference model and why $-\mathcal{L}_{VAE}$ is a proper lower bound.
- From the supplement, it seems a modified version of the bound (MSE instead of gaussian ll and scaled KL term) is actually used (Algorithm 1), again a detail that should be mentioned.
- Is the policy deterministic or stochastic and which loss is used for training? While that information is available in the appendix, I again think it should be stated here for clarity. In particular, the loss, which seems to be down-weighted for the "retrieval" data in the real-world experiment (i.e. there is a factor beta in line 15 of Algorithm 2 in the supplement, which according to table 5 is considerably lower than 1 for the real world data). This seems like an important detail and a method to address the shift between prior and task data.

Questions / further remarks:
- Why is the distance between the means used for ranking during retrieval and not something that considers the standard deviation/variance of the inference distribution?
- Is there any further intuition/ analysis it is sufficient to enforce predictability of the temporal difference (especially as $m_\omega$ seems to be a rather large network for such a presumably simple task) to get the shown performance gains? Some analysis how the learned representations differ would be helpful

Typos:
- "on" missing in line 171
- punctuation missing after equations.

**Quality Of The Limitations Section:**

Limitations are addressed clearly

**Reviewer Expertise:**

4: The reviewer is confident but not absolutely certain that the evaluation is correct

**Robotics Focus:**

Sufficient demonstration on hardware

**Strengths And Weaknesses:**

I think the paper proposes an interesting approach to an important problem in robotics. The main algorithmic contributions, adding the temporal consistency loss and distance-based retrieval of relevant prior data for policy learning are novel and, despite their rather heuristic nature, seem to be effective as shown by the ablation.

In general. the experimental evaluation shows the method's capabilities in several challenging tasks and indicates the method's potential impact. While the presented experiments are sufficient, for an even better assessment of this impact further ablations on the sizes of prior, retrieval, and target data-set would be useful.

While the main ideas are clear and the paper is generally easy to follow, I think there is some rigor missing in the methods section, where the objective is simply stated without context and several details have to be pieced together from the supplement (for more details see issue section)

**Summary Of Recommendation:**

The paper proposes an interesting, novel idea and sufficiently demonstrates the potential impact on a set of challenging experiments. While the key ideas are easy to follow and clear, the paper could be improved with more rigor in the method section

---

> ### Author Response · Authors · 2022-08-26
> **Author Response to Reviewer m8W5 (1/2)**
>
> Dear Reviewer,
>
> Thank you very much for your thorough and constructive review! We appreciate your comment that our contributions are interesting, novel, and have high potential impact. Following your feedback we have added a number of experiments and we address your specific concerns as follows.
>
> > further ablations on the sizes of prior, retrieval, and target data-set would be useful.
>
> This is a great suggestion! Regarding the prior dataset, in our original submission we had ablations on the size and content of the prior dataset (please refer to Appendix B.1). From these results we see that increasing the size of prior data yields greater downstream policy performance, and our method achieves good performance even when the prior dataset does not contain data from the same environment as the target dataset.
>
> We have added a new ablation study on the size of the target dataset. We find for the CALVIN-Setting Up task that increasing the number of target task demonstrations from 30 to 100 yields an increase in success rate from 77.3% to 93.3%. Please refer to Appendix B.3 for additional details and a discussion of the results.
>
> Finally, we have added more detailed ablation studies on the quantity and quality of the retrieved data for policy learning (Appendix B.4). We explore changing the quality of the retrieved data, and also changing the quantity while keeping quality fixed. For the CALVIN-Cleaning Up task we find that the former is important while the latter does have a major impact on performance. CALVIN-Setting Up however offers a different analysis. For this task data quality does not appear to matter. One possible explanation for this observation is that the Setting Up task involves a more diverse range of behaviors than Cleaning Up – the Setting Up task involves manipulating all components of the environment whereas the Cleaning Up task involves a subset. Another potential hypothesis is that the prior data is more biased towards behaviors seen in the Setting Up task. Because many of the behaviors in the Cleaning Up task are mirror behaviors of the Setting Up task, this may result in an unfavorable bias for the Cleaning Up task, necessitating a retrieval procedure to filter out irrelevant behaviors.
>
> The implication of all of these results is that the importance of retrieval may be task and dataset dependent, with some tasks being especially sensitive to the choice of retrieved data.
>
> > there is some rigor missing in the methods section, where the objective is simply stated without context and several details have to be pieced together from the supplement
>
> Thank you for the feedback. Upon revisiting the methods section we also see the concern that many pieces are presented without a consolidated review after presenting these pieces. We believe that Algorithm 1 and Algorithm 2 in the appendix present a comprehensive outline of our method. Due to space constraints we were unable to include these algorithms in the main text. However we have added references to these algorithms in the methods section for interested readers, and have added additional discussion on finer details of our algorithm (MSE reconstruction loss, binary input to the policy, weight factor on policy loss for retrieved dataset).
>
> > VAE loss objective (Equation 1): What exactly is lower bounded (\sum_o^{H-1} log p(a_t | o_t), what are the independence assumptions/graphical models underlying the generative and inference model and why -L_{VAE} is a proper lower bound.
>
> This is a great question. Ajay et al. [1] used nearly an identical underlying VAE objective as ours (the only difference is that they condition the prior on the first observation of the sub-trajectory while we condition on the first and last observation). They show in their analysis that this is a lower bound on the probability of sub-trajectories given the first observation. The same analysis applies to our VAE loss, and we updated the manuscript to reference their theoretical analysis.
>
> [1] OPAL: Offline Primitive Discovery for Accelerating Offline Reinforcement Learning, Ajay et al., ICRL 2021
>
> > From the supplement, it seems a modified version of the bound (MSE instead of gaussian ll and scaled KL term) is actually used (Algorithm 1), again a detail that should be mentioned. Is the policy deterministic or stochastic and which loss is used for training? While that information is available in the appendix, I again think it should be stated here for clarity.
>
> Thank you for pointing out these issues! For the sake of generality we have kept the reconstruction term in Equation 1 stochastic (rather than deterministic MSE loss). However we have added a note right after Equation 1 noting that the skill decoder is deterministic and that we use the MSE for the reconstruction loss.

---

> > ### Author Response · Authors · 2022-08-26
> > **Author Response to Reviewer m8W5 (2/2)**
> >
> > > the loss, which seems to be down-weighted for the "retrieval" data in the real-world experiment (i.e. there is a factor beta in line 15 of Algorithm 2 in the supplement, which according to table 5 is considerably lower than 1 for the real world data).
> >
> > This is an insightful observation. Indeed, for our real robot experiments we believe there is a greater degree of domain shift between the prior and target task datasets, so we chose a more conservative factor compared to the simulation experiments. Due to the high costs of real robot evaluations we did not tune this hyperparameter. We have added additional discussion in the methods section describing the existence and role of this factor (renamed from beta previously now to gamma).
> >
> > > Why is the distance between the means used for ranking during retrieval and not something that considers the standard deviation/variance of the inference distribution?
> >
> > Thank you for the suggestion! We have added an additional experiment that performs retrieval based on KL-divergence distances. Ie. given two inference distributions $q_1$ and $q_2$, we compute their distance as the average forward and reverse KL divergence: $d(q_1, q_2) = \frac{1}{2}(D_{KL}(q_1 || q_2) + D_{KL}(q_2 || q_1))$. This metric effectively incorporates both the mean and standard deviation of the inference distributions. We have added these results to Table 4 in Appendix B.2.
> >
> > We see that KL divergence Retrieval slightly outperforms ours for Setting Up and underperforms Ours for Cleaning Up. Overall there is not a significant difference in the results, and we believe that KL divergence Retrieval is a valid alternative choice to our L2 distance-based retrieval.
> >
> > > Is there any further intuition/ analysis it is sufficient to enforce predictability of the temporal difference (especially as $m_{\mu}$ seems to be a rather large network for such a presumably simple task) to get the shown performance gains? Some analysis how the learned representations differ would be helpful
> >
> > This is a great question. As you suggested, given a high-capacity network it is possible for the temporal consistency model to accurately predict the temporal distance between sub-trajectories without shaping the underlying skill latent representation. However, we found that this prediction task was not trivial, and even with our [128, 128] layer network the temporal prediction distance does not go to 0. This may be due to the high diversity in our datasets, making prediction somewhat challenging.
> >
> > > Typos:
> >
> > Thank you for letting us know! In our updated manuscript we have fixed all of these issues.

---

> > > ### Comment · Reviewer_m8W5 · 2022-08-26
> > > **Response to Authors**
> > >
> > > I thank the authors for their thorough reply and revision. My remaining concerns are addressed and I highly appreciate the added details and comparisons.  Thus, I increase my recommendation to **strong accept** (I can't seem to be able to edit my original review at this point but will do so if that becomes possible after the end of the rebuttal phase)

---

> > > > ### Author Response · Authors · 2022-08-28
> > > > **Thank you!**
> > > >
> > > > We are delighted to hear your positive response, thank you!

---

### Official Review · Reviewer_bYiA · 2022-08-02

**Originality:** Fair
**Technical Quality:** Fair
**Clarity Of Presentation:** Fair
**Impact:** 3

**Recommendation:**

Weak Accept: I recommend accepting the paper, but will not argue for my recommendation if the majority of other reviewers have a different opinion.

**Summary:**

This work proposes an imitation learning framework that extracts temporally-extended sensorimotor skills from prior data and subsequently
learns a policy for the target task with respect to these learned skills.
The contributions of this manuscript are the temporal consistency term in the VAE loss function, a
representational learning objective to enable more predictable and consistent skill representations,and the data retrieval based on the ensuing encoder,
to increase the scope of supervision for the policy.
There are both simulated and real robot results.

**Issues:**

Do a wider range of tasks on the real robot, beyond pick and place.

Improve performance on in the simulation and on the real robot.

LEFTOVERS FROM MAIN REVIEW (MOVED HERE BECAUSE OF STUPID CHARACTER LIMIT)

Small comments on paper:
- sub-trajectories -> trajectory segments - throughout the paper
- 41: pace -> space
- 73: to a specific types of tasks -> to a specific type of tasks
- 93: complimentary -> complementary
- 94: Jargon alert: you are assuming your reader knows what VAE means.
    - spell out (as in line 134) and provide a reference to make your
      paper more accessible.
- 136, 153: Jargon! LSTM has not been explained, spelled out, or had a
  reference provided.
- Equation 2: you might explicitly says that mu() is E() or mean(),
or just use the standard notation E()
- 171: train the policy the aggregate -> train the policy with the aggregate


**Quality Of The Limitations Section:**

Limitations are not well addressed

**Reviewer Expertise:**

4: The reviewer is confident but not absolutely certain that the evaluation is correct

**Robotics Focus:**

Sufficient demonstration on hardware

**Strengths And Weaknesses:**

STRENGTHS

The contributions of this manuscript are the temporal consistency term
in the VAE, and the data retrieval based on the ensuing encoder.

Experiments involve simulation and hardware, and compare the method to
other baselines.
Good set of ablation studies, which ties in nicely with the baseline
 comparison (like against BC-RNN without D_prior, as the authors'
 method without D_prior still performs better)

Data retrieval based on skill matching is interesting

ISSUES and CONCERNS

The real robot tasks could be much more compelling.
The tasks seem limited to pick and place. It would be easy to
use traditional methods to achieve the same results, so why bother with
the complexity and risk of learning approaches?
From a perceptual point of view, for the real robot there is a set of
fixed objects - no variation in size, shape, or appearance. It would have
been better to do the experiments with real vegetables and fruit, which
have natural variation.
There is generalization in terms of adjusting to object and goal locations.
There is no evidence for generalization in terms of objects.
77% performance on these tasks is not impressive.
It is not surprising that the ABC prior data vs.
D task ablation works well (around
line 433) as all the tasks are similar.
What about using the utensils to actually manipulate something else,
pouring the milk,
actually using a real stove (turning knobs and pressing buttons),
spreading butter on toast or something else for the breakfast task,
cutting the bread, sausage, or fish, or
peeling the banana?

This paper presents a simplistic view of what a skill is.
1) Only position, yaw, and gripper state are controlled in the real robot
experiments. This is a reflection of the limited nature of the tasks
explored in this paper.
2) Skills are limited to trajectories (open loop sequence of actions) only, and trajectory-only skills
didn't get robotics very far over the past decades.
That is why robotics  adopted:
 - force control, impedance control, and other control laws
   (including nonlinear control laws)
 - timing and transition rules for finite state machines
 - error handling
In situations with rigid objects and rigid environments, something must be
compliant or infinite (large) forces result. That means either some
mechanical compliance in the robot or some form of force control.
I don't see how the proposed system would be extended to handle these
aspects of skills.

For many tasks a solution has to be a policy u=pi(x). An open loop sequence of
tasks cannot solve unstable tasks, for example.
This is a mistake made in good old fashioned AI.
It is true that demonstrations or any instance of a task execution
are sequences of actions, but this does not imply that a controller
that just emits previously successful sequences of actions will succeed.
Consider riding a bicycle. A bicycle rider can't just play back the
steering and balance motions used in a previous ride, they will quickly
fall over.

I had to go to the supplementary document to figure out what the
actions were. It turned out they were different in the 3 cases:
452: agent operates the robot via joint torque control resulting in a
9-dimensional action space
(joint torque control is probably a bad choice)
474: agent operates the robot via inverse kinematics control resulting
in a 7-dimensional action space.
507: OSC, position and yaw of the end effector, gripper state.

It is not clear what the gripper state is (open/close, or finger
positions).
How is gripper closing controlled? Do the gripper fingers
go to a fixed position, or is a closing force specified?
If positions are controlled, what happens if a position is too small
for the object being gripped? The gripper outputs a maximum force?
All of this should be clearly explained in the paper itself.

It is not clear how goals are specified to the system? Desired final
positions for objects? An optimization criterion? Some measure of
similarity to a demonstration?

The network architectures chosen were not justified or compared to
alternative architectures.

The paper does not make clear which trajectory segments (sub-trajectories)
are used from D_target or D_prior. Are all possible trajectory segments
used, in which case adjacent segments of length H share H-1 values
(which is implied by the use of the term "all" in line 160 and 162)?
Or are trajectory segments distinct, with no overlap between adjacent
segments?
Or are trajectory segments initiated or terminated at significant events
like making or breaking contacts?
Prior work (6, 8) is described as "parsing" the demonstrations (line 122),
but how does the parse/segmentation work in this paper?

In line 163 similar sub-trajectories are retrieved. If sub-trajectories
overlap, this will choose many from the same demonstration. Is that good
or bad?

How is H picked? I would imagine it would have a huge possible range
(some skills are quite short, while others (like steering a car) last
a long time (as long as one is driving the car)).

It is not obvious to this reader that predicting temporal differences
is useful, but let's see what happens in the ablations section.
One could always decode the latent representations producing trajectory
segments and then explicitly correlate the trajectory segments.
How is that a constraint on the latent representation, if the
encoder/decoder pair already work well?

Is F, the number of observations in the argument to the policy, a constant?
How it chosen? Wouldn't it make more sense to map observations into a
observation-latent-space (this is what state estimators like Kalman
filters do), and use the latent variable (the "estimated state") as
the first component of the training points as well as the index into
the policy? Perhaps the output of the perception network is such an
observation latent space.
[I see this is addressed starting at line 547 in the supplementary materials.
Perhaps it should be discussed in the paper.]

How is it decided where to start the readout of a trajectory? Always
at the start?

How is it decided when a new/different skill should be executed?
When the readout of the trajectory is finished (which is implied by
the paper's reference to "H timesteps" on line 159)?
There is no preemption of the current trajectory if things do
not go as expected? This leads to robots doing stupid things like trying
to close an automatic door when the person going through is having trouble
with their luggage, and the door starts swatting them.

In the discussion of "retrieval-based", what is retrieved?
The actual trajectory segment? The latent representation?

The policy is trained using instances (input: o_fs, output: tau).
The retrieval-based policy learning focuses on similarities of tau.
Why not focus on similarities of o_fs? It would seem to this reader
that for a policy similar inputs should lead to similar outputs
(except at a policy discontinuity), but I would not expect that
similar outputs are caused by similar inputs (Many (most?) policies
are many to one mappings, in that the dimension of the state or
observation space is usually much larger than the dimension of the
action space, and by replacing actions with skills, the dimension
of the policy output space is reduced further.)

I don't understand the purpose of the binary input bit saying whether
the training point is from the prior or target demonstrations.

How are initial state and goal conditioning of trajectories done (how
are the trajectories morphed to start and stop in the right places)?
How are grasping trajectories modified to be appropriate for an object.

The real robot video is impressive. I wouldn't bother presenting the
simulation results, and just focus on the real results.  You give 9
lines to real results, and more than 10 times that to simulation
results. Do the comparisons and ablations on the real robot!  Much
more convincing!  Though simulation is helpful to speed up
development and evaluation, the reviewer believes that
it is ultimately the real robot experiment that demonstrates the value
of the authors' method. The reviewer would suggest the authors put
more emphasis on the real robot experiment and conduct quantitative
analyses similar to the ones completed in simulation on the real robot
setup. This change could make the readers better appreciate the value
of the authors' method and its performance.

The authors have used only "success rate" as the performance metric in
the real robot experiment.  However, for the authors' method to be
practical in the real world, the reviewer believes that performance
metrics are not limited to success rate---other, even more important,
performance metrics exist.  Since the task in the real robot
experiment is setting up a breakfast table and cooking a meal, one of
the relevant performance metrics could be the total time for
completing the entire meal preparation process.  Obviously, if such a
robot cooks a meal in a real household, the users would tend to prefer
the robot the finish the meal within a reasonable and predictable
amount of time. In this regard, the reviewer would suggest the authors
to put more emphasis on practical performance metrics for the task
chosen for the real robot experiment, instead of a single "success
rate".

How do we know whether the skills failed (the output action sequence
is not good enough)? It appears the simulators simulate physics as joint
torques are generated, but there is little description of them.
In lines 577-584 the words "we evaluate" are used, but I have no
idea what those words actually mean. What is a checkpoint? Who chose
them? How are they evaluated?

It’s not clear exactly how this paper fits into the broader context of
existing research in the field. A major shortcoming is the discussion
of prior approaches for skill learning - the authors seem to only be
referencing works that use variational auto-encoders.  While the
proposed method does demonstrate relatively good performance in the
evaluation domains, the discussion of the results does not provide a
lot of intuition about why other baselines failed to perform as
well. I think this paper would be much improved by more explicitly
describing how this work compares to prior works and the
baselines. Specifically, what mechanisms can we attribute to the
success of the method?

The distinction between a skill and a policy is confusing. The terms
skill and policy are used interchangeably in many contexts. I think by
skills the authors are referring to the latent embedding.

The authors claim “A major limitation of existing approaches is that
they only focus on using the prior data for skill learning but not
policy learning.” This needs to be better supported. Among works that
don’t make a distinction between “skill” learning (i.e. learning
latent embeddings) and policy learning, prior data has been used to
update the
policy. (http://proceedings.mlr.press/v139/wulfmeier21a/wulfmeier21a.pdf
is just one example)

Figure 2: I think the components of the system should be better
justified. Why make a distinction between skill learning and policy
learning? Why not just learn low level policies that are “skills” and
a high-level policy that selects the skills? What is the advantage of
continuous skill embeddings vs. having a discrete high-level policy
that learns to select skills?

Section 5.1: Quality of the prior and target datasets - Because
imitation learning is used and there is no explicit representation of
task goals or reward, the quality of learned policy is dependent on
the quality of the data in the prior and target datasets. It would be
helpful to have more detail about how each dataset affects the learned
policy. For instance, I would imagine that it is more important for
the target dataset to have high quality demonstrations, since the
samples from the prior dataset are chosen to be close to the
target. What would happen if the target had poor demonstrations and
but the prior had higher quality demonstrations and vice versa?

What is the relationship between the number of sub-trajectories
sampled from the prior dataset and policy performance? What would
happen if you thresholded the distance instead of selecting a fixed
number? If the prior was poor quality or not particularly relevant to
the target, it would probably be more beneficial to use threshold for
selecting sub-trajectories. Depending on the data, it may be better to
not take any samples from the prior.

Line 225: What is the form of the non-parametric policy used in the
FIST baseline? And what would the performance of this method be if
data retrieval was used with the non-parametric policy?

How could knowledge of task goal/reward be incorporated into this
framework? It seems like indexing the most similar trajectories would
not always be sufficient to filter out bad examples. There can be poor
demonstrations that are closer to the target than other samples, but
still would not result in reaching the goal state. Relabeling the
sub-trajectories with rewards for the target task is a common approach
used in multi-task reinforcement learning. Is there a benefit to
framing this as an imitation learning problem without using task
reward signals?

Every single task needs re-learning: there's no obvious way of
changing the order of parts of the task for example: if the agent has
learnt to do A-->B-->C, it has to re-learn to do C-->B-->A. (that
said, if I open a cabinet drawer, I can take a fork then a knife, or
take a knife then a fork without a problem. But if I learn to play a
music sheet, even just a few bars, I can't directly play it in
reverse. What is the fundamental difference? and if I learn to play it
in reverse, there's probably going to be catastrophic forgetting for
the normal play)

The temporal consistency for skill embedding seems only suitable for
restricted domains with highly repetitive and ordered action
sequences. Not really suitable for improvisation and
problem-solving. Discuss.

The distinction between a skill and a policy is confusing. The terms
skill and policy are used interchangeably in many contexts. I think by
skills the authors are referring to the latent embedding.

Comments on the experiments:
- for the baselines: any training on D_target alone is unfair as
  only the target task dataset is used. That said, It makes more sense
  after the ablation study when comparing to the "non-retrieval"
  study.
- What is meant by "struggles to find appropriate goal observations to reach"\
 ?
- Why does FIST perform better for no-microwave?
- Could the authors provide a comment on why BC-RNN (FT) performs so poorly o\
n hardware experiments?
- There is no real inherent generalizability, the training and
  execution domains are the same. The no-microwave in D_prior was
  interesting though, showing that the encoder still made enough sense
  of this new skill to supervise the policy learning. However, other
  methods also seem pretty robust against no-microwave.

**Summary Of Recommendation:**

I would like to see more challenging tasks done on the real robot.

Performance on the tasks in simulation (77%) was not good.

I would like to see a richer notion of what a skill is, ideally a policy mapping low level states to low level actions.

Since information on whether trajectory segments was not clear, it is difficult to assess the role of temporal consistency.

I would focus on real robot results.

POST REBUTTAL COMMENT

I am still in favor of accepting this paper.

---

> ### Author Response · Authors · 2022-08-26
> **Author Response to Reviewer bYiA (1/8)**
>
> Dear Reviewer,
>
> Thank you very much for taking the time to carefully review our paper! We are glad that you appreciate our ablation studies and find our data retrieval procedure interesting. We found your feedback extremely valuable and constructive, and we have made a number of changes to our paper following your suggestions. We address your specific concerns below.
>
> > What about using the utensils to actually manipulate something else, pouring the milk, actually using a real stove (turning knobs and pressing buttons), spreading butter on toast or something else for the breakfast task, cutting the bread, sausage, or fish, or peeling the banana?....The reviewer would suggest the authors put more emphasis on the real robot experiment and conduct quantitative analyses similar to the ones completed in simulation on the real robot setup.
>
> Our ultimate goal is to deploy physical robots in challenging real world settings, so we certainly share your sentiment! This is in fact the reason why we conducted real world experiments – we wanted to demonstrate that our method has potential to solve practical real world problems. While it would be ideal to run comprehensive comparisons on more interesting tasks, the cost of real world evaluation is a significant bottleneck that precludes us from running large-scale experiments. For reference, due to the low cost of evaluation in simulation we were able to evaluate 6000 episodes for each baseline (per task). In contrast, in the real world this volume of evaluations would likely take weeks as it demands a human operator to continually oversee the robot and manually reset the scene for each evaluation. Due to the limited budget of evaluation, we instead opted to run a limited but thorough set of evaluation evaluations for the most promising methods from our simulation results, in this case our method and BC-RNN (FT).
>
> Nonetheless, we have taken the time to run additional real world experiments, including one new ablation result and one new task. To see the value of prior data we ran the No Prior ablation for the cook task, achieving 53.3% success rate compared to our 73.3%. Interestingly we see that the No Prior ablation significantly outperforms the BC-RNN (FT) baseline on this task (23.3%), even though the latter additionally has access to prior data. We also include results for a new task which we call “setup and cook”. This task is similar to the original cook task, except that the frying pan is initialized on the table (instead of the stove) and the agent must place the pan onto the stove before placing the fish and tomato into the pan. We provide additional details about this task in Appendix C.3. Note that manipulating the pan is more challenging than manipulating the food items, as the agent must be careful not to tip the pan or knock over the other objects as it is transporting the pan to the stove. In our evaluations we find that our method achieves a success rate of 76.7% while BC-RNN (FT) achieves 46.7%.
>
> > There is generalization in terms of adjusting to object and goal locations. There is no evidence for generalization in terms of objects. 77% performance on these tasks is not impressive.
>
> We agree that our empirical evaluations concern limited forms of generalization, and we have updated the Limitations section of our manuscript to reflect this. We note that while we did not study generalization in terms of objects, generalization to new object positions is still a very challenging problem due to the small number of target task demonstrations. Under this setting we show that our method significantly outperforms a number of existing methods.
>
> > Only position, yaw, and gripper state are controlled in the real robot experiments. This is a reflection of the limited nature of the tasks explored in this paper.
>
> This is indeed the case. For the real world tasks that we considered we did not believe roll and pitch actuation would be necessary and we opted for a simpler action space. While the action space in our real world experiments is simplified compared to simulation, our real world experiments considered a larger number of free-moving objects and examined greater variability in the initial placement of objects.

---

> > ### Author Response · Authors · 2022-08-26
> > **Author Response to Reviewer bYiA (2/8)**
> >
> > > For many tasks a solution has to be a policy u=pi(x). An open loop sequence of tasks cannot solve unstable tasks, for example….Consider riding a bicycle. A bicycle rider can't just play back the steering and balance motions used in a previous ride, they will quickly fall over.
> >
> > We would like to clarify that both our policy and skill models are **closed-loop** models that operate in a hierarchical fashion. At the high level the policy $\pi(o_{fs} , id = 0)$ takes in observations and outputs a skill embedding z. At the low level we unroll the the skill decoder RNN $p_{\psi}(o_t, z)$ for H timesteps, where for each step t it takes in the skill embedding and observation $o_t$ as inputs and outputs actions. Please note that since the skill decoder takes observations $o_t$ as input it is also a closed-loop module.
> >
> > > I had to go to the supplementary document to figure out what the actions were. It turned out they were different in the 3 cases
> >
> > Thank you for making this observation. Indeed our experiments operate on a variety of robot controllers. Since we used existing datasets for the Franka Kitchen and CALVIN domains we were confined to the robot controllers that these datasets came with. For our real world experiments we had the liberty of choosing our own robot controller. We found operational space control (OSC) to be a fitting choice, as it offers task-space compliant behavior that makes for a more intuitive data collection experience.
> >
> > > It is not clear what the gripper state is (open/close, or finger positions). How is gripper closing controlled? Do the gripper fingers go to a fixed position, or is a closing force specified? If positions are controlled, what happens if a position is too small for the object being gripped? The gripper outputs a maximum force?
> >
> > The gripper state is either the position of the gripper fingers (Franka Kitchen, Real World Kitchen) or the opening width of the gripper (CALVIN). The gripper action is a continuous 1-D variable, and we interpret this as either opening (if < 0) or closing (if > 0). The gripper is controlled via position control. When the agent specifies a closing action the position target of the controller is set to close the gripper fingers all the way (and for opening the target is set to open the gripper fingers all the way). There are limits on the force and velocity of the fingers in order to ensure gripper stability. We have updated the manuscript with these details (Appendix D.5).
> >
> > > It is not clear how goals are specified to the system? Desired final positions for objects? An optimization criterion? Some measure of similarity to a demonstration?
> >
> > Thank you for the question. The notion of “goal” can encompass various levels of abstraction, eg. a type of task to perform, or more specific information such as the exact desired position of objects. The target task demonstrations in practice convey the high-level goal of the task, and through our imitation learning algorithm we train our policy to satisfy this goal. However, for a given high-level task there may be more specific low-level goals that we want to agent to satisfy, eg. through a goal image. Our standard method does not take this low-level information into account, as the tasks that we considered do not have a high degree of variability in their final desired outcome. However, we can modify our policy to explicitly take this information into account by conditioning the policy network on additional context information.
> >
> > > The network architectures chosen were not justified or compared to alternative architectures.
> >
> > Our implementation is based on top of robomimic [1], a recent open source framework with extensive benchmarking results across a number of imitation learning algorithms. We adopted the same neural modules (same RNN backbone, VAE, visual perception encoders) for our algorithm, and in fact our BC-RNN baseline uses the exact implementation from robomimic.
> >
> > [1] What Matters in Learning from Offline Human Demonstrations for Robot Manipulation, Mandlekar et al., CoRL 2021.

---

> > > ### Author Response · Authors · 2022-08-26
> > > **Author Response to Reviewer bYiA (3/8)**
> > >
> > > > The paper does not make clear which trajectory segments (sub-trajectories) are used from D_target or D_prior. Are all possible trajectory segments used, in which case adjacent segments of length H share H-1 values (which is implied by the use of the term "all" in line 160 and 162)? Or are trajectory segments distinct, with no overlap between adjacent segments?
> > >
> > > This is a great question. For simplicity we sample batches from all possible trajectories segments to train on. This means there can be overlap between the sampled segments. We have updated the paper to clarify this detail (see Appendix D.2). We did so for simplicity, and also in following the same convention as related work [1, 2, 3, 4]. However, parsing demonstrations at distinct events has potential to induce more semantically meaningful skills, as is an interesting avenue for future work.
> > >
> > > [1] Accelerating Reinforcement Learning with Learned Skill Priors, Pertsch et al., CoRL 2020
> > >
> > > [2] Demonstration-Guided Reinforcement Learning with Learned Skills, Pertsch et al., CoRL 2021
> > >
> > > [3] OPAL: Offline Primitive Discovery for Accelerating Offline Reinforcement Learning, Ajay et al., ICRL 2021
> > >
> > > [4] Hierarchical few-shot imitation with skill transition models, Hakhamaneshi et al., ICLR 2022
> > >
> > > > In line 163 similar sub-trajectories are retrieved. If sub-trajectories overlap, this will choose many from the same demonstration. Is that good or bad?
> > >
> > > This is an insightful observation. Retrieving numerous sub-trajectories from the same demonstration should not be a concern unless this significantly limits the diversity of the retrieved sub-trajectories (eg. most come from the same demonstration and few sub-trajectoires are retrieved from other demonstrations). We can add additional constraints on our retrieval algorithm to limit the relative proportion of sub-trajectories retrieved from any one demonstration. A more basic alternative is to retrieve more sub-trajectories to encompass a larger number of demonstrations. For simplicity of implementation we think the latter may be a good solution.
> > >
> > > > How is H picked? I would imagine it would have a huge possible range (some skills are quite short, while others (like steering a car) last a long time (as long as one is driving the car)).
> > >
> > > In our earlier experiments we explored a range of options for H (1, 5, 10, 25), and found H=10 to work the best. We note that a number of related works [1, 2, 3, 4] also use H=10. Longer horizons may be helpful in some settings, however we hypothesize that RNN-based architectures lack the capacity to accurately predict actions over significantly longer horizons. It would be interesting to investigate if the optimal H changes under a Transformer-based [5] architecture.
> > >
> > > [1] Accelerating Reinforcement Learning with Learned Skill Priors, Pertsch et al., CoRL 2020
> > >
> > > [2] Demonstration-Guided Reinforcement Learning with Learned Skills, Pertsch et al., CoRL 2021
> > >
> > > [3] OPAL: Offline Primitive Discovery for Accelerating Offline Reinforcement Learning, Ajay et al., ICRL 2021
> > >
> > > [4] Hierarchical few-shot imitation with skill transition models, Hakhamaneshi et al., ICLR 2022
> > >
> > > [5] Attention is all you need, Vaswani et al., NeurIPS 2017
> > >
> > > > One could always decode the latent representations producing trajectory segments and then explicitly correlate the trajectory segments. How is that a constraint on the latent representation, if the encoder/decoder pair already work well?
> > >
> > > While it is possible to achieve low reconstruction error with a simple autoencoder skill model, it is not necessarily the case that this is a semantically meaningful representation suitable for downstream control. Ideally we would like a representation that has both low reconstruction error and is predictable and consistent. The VAE prior is one step towards learning a more predictable presentation, and our temporal consistency objective is yet another step towards this goal. These auxiliary objectives can complement the original reconstruction object well, resulting in a more consistent representation without necessarily degrading reconstruction quality.
> > >
> > > > Is F, the number of observations in the argument to the policy, a constant? How it chosen? Wouldn't it make more sense to map observations into a observation-latent-space
> > >
> > > In all of our experiments we set F to a constant value of 10, following the same setting from BC-RNN in robomimic [1]. Our policy network is identical to that of BC-RNN – the network (1) takes F observations as input, (2) encodes these F observations individually using the observation encoder, and then (3) processes these encoded observations into one unified representation with a Long Short Term Memory (LSTM) network. Please refer to the “Policy” subsection in Appendix D.1 for additional details.
> > >
> > > [1] What Matters in Learning from Offline Human Demonstrations for Robot Manipulation, Mandlekar et al., CoRL 2021

---

> > > > ### Author Response · Authors · 2022-08-26
> > > > **Author Response to Reviewer bYiA (4/8)**
> > > >
> > > > > How is it decided where to start the readout of a trajectory? Always at the start?
> > > >
> > > > We sample sub-trajectories uniformly at random from the prior and target datasets to train our model. More specifically, for each dataset we concatenate all trajectories into one continuous stream of data and uniformly sample sub-trajectories from this stream. For training the policy we additionally train on the frame stack of observations preceding the sampled sub-trajectory. There are some edge cases, such as when the sub-trajectory intersects with the next trajectory and when the frame-stack intersects with the previous trajectory. We deal with these cases by padding all data from the offending consecutive trajectory with the first / last observation of the current trajectory.
> > > >
> > > > > How is it decided when a new/different skill should be executed? When the readout of the trajectory is finished (which is implied by the paper's reference to "H timesteps" on line 159)? There is no preemption of the current trajectory if things do not go as expected?
> > > >
> > > > Thank you for the question. For simplicity we do not preempt skill execution, ie. we execute each skill for H timesetps and return control afterwards to the policy to select a new skill. While this may cause unintended consequences we think these concerns are alleviated for two reasons. (1) The skill model is a closed loop controller, and it can (to an extent) react to current environment conditions during skill execution. (2) Given the context of an entire episode (hundreds of timesteps long) the skill horizon H is relatively short (just 10 timesteps); thus even if there are erroneous behaviors during skill execution, the policy is still operating at high frequency and can react accordingly.
> > > >
> > > > > In the discussion of "retrieval-based", what is retrieved? The actual trajectory segment? The latent representation?
> > > >
> > > > The role of our retrieval process is to expand the scope of training data for the policy. The policy training data consists of (input, output) pairs, where the input is a frame stack of observations and output is a skill latent representing the skill that the policy should select given the frame stack observation input. Thus, our retrieval process is retrieving both the frame stack of observations (input) and the latent representation z (output) corresponding to the H-step sub-trajectory immediately following the frame stack of observations.
> > > >
> > > > > The policy is trained using instances (input: o_fs, output: tau). The retrieval-based policy learning focuses on similarities of tau. Why not focus on similarities of o_fs?
> > > >
> > > > This is an insightful observation. Interestingly we had a different perspective on the many-to-one mapping point. Due to the highly exploratory nature of the prior data, the dataset exhibits a high degree of mult-modality, resulting in many possible outputs (tau) given the same input (o_fs). For example, upon picking up an object the human collector sometimes proceeded to placing the object in the drawer, placing the object in the cubby, or stacking the object on top of another object. Thus, retrieving on the basis of similarities in o_fs will likely result in retrieving a very diverse set of outputs (tau) that may not necessarily be relevant to the target task. However we share the perspective that retrieving based on inputs o_fs is promising, and one idea is to retrieve based on similarity to *both* o_fs and tau.
> > > >
> > > > > I don't understand the purpose of the binary input bit saying whether the training point is from the prior or target demonstrations.
> > > >
> > > > While the purpose of the retrieval process is to expand the scope of training data for the policy, the retrieved data may not always be entirely relevant to the target task. Even if the retrieval data is highly relevant, it may make the training distribution more multi-modal which can lead to optimization difficulties. We therefore use the binary bit to minimize the effect of these factors. In effect, we can view the retrieved data as a means to regularize the policy rather than a direct augmentation procedure.
> > > >
> > > > > How are initial state and goal conditioning of trajectories done (how are the trajectories morphed to start and stop in the right places)? How are grasping trajectories modified to be appropriate for an object.
> > > >
> > > > We do not explicitly retarget or morph demonstrations to learn target tasks. Instead we distill the diverse sub-trajectories in the prior dataset into the skill model, and we use this model to generalize to new, unseen sub-trajectories for downstream target tasks. Note that we additionally fine-tune the skill model on sub-trajectories in the target task demonstrations (see Algorithm 2 in Appendix) in case the prior dataset does not sufficiently capture sub-trajectories seen in the target dataset.

---

> > > > > ### Author Response · Authors · 2022-08-26
> > > > > **Author Response to Reviewer bYiA (5/8)**
> > > > >
> > > > > > The reviewer would suggest the authors to put more emphasis on practical performance metrics for the task chosen for the real robot experiment, instead of a single "success rate".
> > > > >
> > > > > This is a great point! While we believe that success rate is the most informative metric of the agent’s performance, other useful metrics exist, such as the time to complete the task. Per your suggestion we were hoping to retroactively measure other metrics, but at the time of our submission we unfortunately did not save detailed logs of all of the evaluations. This is a good lesson to save logs for all future real world evaluations.
> > > > >
> > > > > > How do we know whether the skills failed (the output action sequence is not good enough)? It appears the simulators simulate physics as joint torques are generated, but there is little description of them. In lines 577-584 the words "we evaluate" are used, but I have no idea what those words actually mean. What is a checkpoint? Who chose them? How are they evaluated?
> > > > >
> > > > > We follow an identical evaluation protocol as Mandlekar et al. [1]. We elaborate on the details as follows. For our simulation experiments we average results over 3 seeds. For each seed, we consider the checkpoint with the highest success rate – each checkpoint refers to the agent at a specific epoch during the training process. Note that we equally space checkpoints across a fixed epoch frequency during training (please refer to Table 6 in the appendix for more details). To evaluate a specific checkpoint, we measure the average success rate across multiple episodes. We run each episode either until the task is solved (success) or until the maximum number of episode steps have been reached (failure). We have detailed in Appendix section C how we specifically qualify success for each of our tasks.
> > > > >
> > > > > [1] What Matters in Learning from Offline Human Demonstrations for Robot Manipulation, Mandlekar et al., CoRL 2021.
> > > > >
> > > > > > It’s not clear exactly how this paper fits into the broader context of existing research in the field. A major shortcoming is the discussion of prior approaches for skill learning - the authors seem to only be referencing works that use variational auto-encoders.
> > > > >
> > > > > Thank you for the feedback. We have updated our related works section with a detailed discussion on skill-based imitation learning. We discuss works that segment demonstrations into variable-length segments using variational autoencoders, Bayesian online changepoint detection, and sequence alignment, among other approaches. We additionally discuss how our work fits within the context of these prior approaches.
> > > > >
> > > > > > I think this paper would be much improved by more explicitly describing how this work compares to prior works and the baselines. Specifically, what mechanisms can we attribute to the success of the method?
> > > > >
> > > > > We attribute the success of our method to two primary factors:
> > > > >
> > > > > (1) First, our method benefits from **hierarchical abstractions** provided by our skill representation, enabling us to significantly outperform methods that use a flat policy. We can see this from the No Prior Data ablation compared to the BC-RNN baseline. Even though both methods use the same underlying RNN policies and are trained on the same data (the target dataset), our No Prior Data ablation outperforms BC-RNN by over 15% on the CALVIN tasks. This result is in line with a large body of recent work showing the benefits of hierarchy in learning policies [1, 2, 3, 4, 5]. Compared to these works, we highlight (through our No TC ablation) our specific contribution in learning a more predictable and consistent skill representation, yielding better performance.
> > > > >
> > > > > (2) Second, our method is able to effectively leverage prior robotic data along **all levels of the hierarchy** to learn robust policies for target tasks. The most closely related prior work to ours is OPAL, which only uses the prior data for learning the low-level skills. Note that our method without retrieval falls back to the imitation learning variant of OPAL. We highlight in our ablations (No Retrieval) the importance of leveraging the rich supervision from the prior dataset to train the high-level policy.
> > > > >
> > > > > In sum these factors go hand in hand. Effective hierarchical abstractions, coupled with mechanisms that effectively leverage the prior data, allow us to achieve strong results.
> > > > >
> > > > > [1] Relay Policy Learning: Solving Long-Horizon Tasks via Imitation and Reinforcement Learning, Gupta et al., CoRL 2019
> > > > >
> > > > > [2] Self-Consistent Trajectory Autoencoder: Hierarchical Reinforcement Learning with Trajectory Embeddings, Co-Reyes et al., ICML 2018
> > > > >
> > > > > [3] GTI: Learning to Generalize Across Long-Horizon Tasks from Human Demonstrations, Mandlekar et al., RSS 2020
> > > > >
> > > > > [4] Accelerating Reinforcement Learning with Learned Skill Priors, Pertsch et al., CoRL 2020
> > > > >
> > > > > [5] OPAL: Offline Primitive Discovery for Accelerating Offline Reinforcement Learning, Ajay et al., ICRL 2021

---

> > > > > > ### Author Response · Authors · 2022-08-26
> > > > > > **Author Response to Reviewer bYiA (6/8)**
> > > > > >
> > > > > > > The authors claim “A major limitation of existing approaches is that they only focus on using the prior data for skill learning but not policy learning.” This needs to be better supported. Among works that don’t make a distinction between “skill” learning (i.e. learning latent embeddings) and policy learning, prior data has been used to update the policy
> > > > > >
> > > > > > Thank you for this careful observation! You are certainly correct that there are works that do not make this distinction between skills and policies and train a policy directly from prior data and task data (similar to our BC-RNN (FT) baseline). We have revised the text to underscore this distinction.
> > > > > >
> > > > > > > The distinction between a skill and a policy is confusing. The terms skill and policy are used interchangeably in many contexts. I think by skills the authors are referring to the latent embedding….Figure 2: I think the components of the system should be better justified. Why make a distinction between skill learning and policy learning? Why not just learn low level policies that are “skills” and a high-level policy that selects the skills?
> > > > > >
> > > > > > There are a few important distinctions between our skill learning and policy learning procedures. (1) Different training data: we train our skills on the target task demonstrations and **all** of the prior data, while we train our policy on the target task demonstrations and a **retrieved subset** of the prior data. (2) Different architectures and training objectives: our skill model is a VAE-based model that learns a latent space of behaviors with auxiliary objectives to ensure a well-structured latent space, while our policy involves a simple RNN model and a simple behavior cloning loss. (3) Different roles in decision-making: our skills serve to fulfill short-horizon behaviors while our policy reasons about long-horizon tasks.
> > > > > >
> > > > > > As a result of these differences we opted to give distinct names to these two components to avoid potential confusion between the two. However, we understand that the term skill can be overloaded to refer to different concepts – either the skill embedding or the skill decoder. We have updated the manuscript to explicitly differentiate between these concepts to improve the readability of the paper.
> > > > > >
> > > > > > > What is the advantage of continuous skill embeddings vs. having a discrete high-level policy that learns to select skills?
> > > > > >
> > > > > > We believe that a continuous skill embedding space offers an expressive representation of behaviors, allowing for the agent to interpolate between behaviors and synthesize new ones. Discrete representations can also express a wide range of behaviors given a sufficiently high cardinality, however learning a policy to select over such a high number of discrete skills may present optimization challenges. An interesting avenue for future work would be to consider a hybrid skill embedding space with discrete and continuous components. Under this embedding space the discrete variables serve as “modes” that represent a class of behavior (pushing, opening, etc), while the continuous variables further specify how to instantiate a behavior (where to push to, how much to open a door, etc).
> > > > > >
> > > > > > > Section 5.1: It would be helpful to have more detail about how each dataset affects the learned policy. For instance, I would imagine that it is more important for the target dataset to have high quality demonstrations, since the samples from the prior dataset are chosen to be close to the target. What would happen if the target had poor demonstrations and but the prior had higher quality demonstrations and vice versa?
> > > > > >
> > > > > > One limitation of imitation learning algorithms is that they often demand a large number of high-quality demonstrations in order to perform well [1]. Providing a large number of high-quality target task demonstrations can be very burdensome, but we believe that providing a relatively small number (30) is reasonable. For reference for our target tasks we only put forth 30 minutes towards collecting reasonably high-quality demonstrations. A very interesting question is how the quality of the prior dataset can affect learning on downstream target tasks. Unfortunately we did not have control over the quality of the prior datasets in simulation as they were collected by other human demonstrators. Our hypothesis is that the quality of the prior data does matter, however this remains to be validated once we gain access to prior datasets with varying degrees of suboptimality.
> > > > > >
> > > > > > [1] What Matters in Learning from Offline Human Demonstrations for Robot Manipulation, Mandlekar et al., CoRL 2021.

---

> > > > > > > ### Author Response · Authors · 2022-08-26
> > > > > > > **Author Response to Reviewer bYiA (7/8)**
> > > > > > >
> > > > > > > > What is the relationship between the number of sub-trajectories sampled from the prior dataset and policy performance? What would happen if you thresholded the distance instead of selecting a fixed number?
> > > > > > >
> > > > > > > Capping the retrieval distances according to a distance threshold is a promising idea. Our latest implementation has both options available (either setting a distance threshold or a percentage to retrieve). We will leave the decision of which retrieval mechanism to use to users of our algorithm.
> > > > > > >
> > > > > > > > Line 225: What is the form of the non-parametric policy used in the FIST baseline? And what would the performance of this method be if data retrieval was used with the non-parametric policy?
> > > > > > >
> > > > > > > The non-parametric (** better named semi-parametric **) policy involves a learned distance model that differentiates between a pair of observations H steps apart (positives) and random pairs observations (negatives). We train the distance model jointly on both the target task demonstration dataset and the entire prior dataset (specifically we sample a batch from each dataset and merge into one joint batch). Because the distance model is a general model that discriminates between pairs of observations and is not tied to any specific target task, we use the entire prior dataset rather than a retrieved subset.
> > > > > > >
> > > > > > > Given this distance model, FIST uses a semi-parametric policy to localize to a goal state in the target task demonstration dataset and subsequently uses an inverse dynamics model to reach this goal state. In our FIST implementation we use the same underlying skill model as our method (same encoder, decoder, inverse dynamics VAE prior, etc) to reach goals. We refer readers to Hakhamaneshi et al. [1] for a more detailed description of FIST and our codebase (see algo/fist.py) for our FIST implementation.
> > > > > > >
> > > > > > > ** Please note: technically the FIST policy is a semi-parametric model as it involves a learned distance model. We have modified the manuscript to refer to FIST as a semi-parametric method. **
> > > > > > >
> > > > > > > [1] Hierarchical few-shot imitation with skill transition models, Hakhamaneshi et al., ICLR 2022
> > > > > > >
> > > > > > > > How could knowledge of task goal/reward be incorporated into this framework?...Relabeling the sub-trajectories with rewards for the target task is a common approach used in multi-task reinforcement learning. Is there a benefit to framing this as an imitation learning problem without using task reward signals?
> > > > > > >
> > > > > > > Prior work skill-based reinforcement learning approaches [1, 2, 3] have proposed methods to learn high-level policies using task rewards with skills serving as the underlying action space of the policy. The most closely related work is OPAL [3] which uses a very similar skill learning procedure as our method and offline reinforcement learning on the target task demonstrations to train the high-level policy. Due to the burden of obtaining target task reward labels on the prior dataset, OPAL only uses the target task demonstrations to train the high-level policy. In contrast, as our method does not require any reward labels, we can additionally benefit from additional supervision from the prior dataset to train the policy. However, we can in principle consider a reward-based reinforcement learning variant of our method that uses unlabeled data sharing (UDS) [4] to train the policy. It would be interesting to see how this variant compares to our standard behavior cloning policy learning procedure.
> > > > > > >
> > > > > > > [1] Accelerating Reinforcement Learning with Learned Skill Priors, Pertsch et al., CoRL 2020
> > > > > > >
> > > > > > > [2] Demonstration-Guided Reinforcement Learning with Learned Skills, Pertsch et al., CoRL 2021
> > > > > > >
> > > > > > > [3] OPAL: Offline Primitive Discovery for Accelerating Offline Reinforcement Learning, Ajay et al., ICRL 2021
> > > > > > >
> > > > > > > [4] How to Leverage Unlabeled Data in Offline Reinforcement Learning, Yu et al., ICML 2022
> > > > > > >
> > > > > > > > Every single task needs re-learning: there's no obvious way of changing the order of parts of the task for example: if the agent has learnt to do A-->B-->C, it has to re-learn to do C-->B-->A.
> > > > > > >
> > > > > > > We agree that compositional reasoning – the ability to compose existing behaviors to learn new tasks quickly – is a highly desirable property for intelligent agents. Our framework encompasses this compositional reasoning to an extent, as we can re-use our skill model to solve different downstream target tasks. More concretely in our implementation we first pre-train the skill model on the prior dataset (see Algorithm 1 in Appendix) and subsequently use and fine-tune the skill model to learn target tasks (see Algorithm 2). In this sense we can learn new tasks more efficiently without learning the skills from scratch – we simply fine-tune the skills on the target task demonstrations. For future work we can investigate additionally pre-training a general policy on the prior data and fine-tuning this policy for a given target task, similar to the BC-RNN (FT) baseline.

---

> > > > > > > > ### Author Response · Authors · 2022-08-26
> > > > > > > > **Author Response to Reviewer bYiA (8/8)**
> > > > > > > >
> > > > > > > > > The temporal consistency for skill embedding seems only suitable for restricted domains with highly repetitive and ordered action sequences. Not really suitable for improvisation and problem-solving. Discuss.
> > > > > > > >
> > > > > > > > We would like to clarify the contribution of our work. Our ablation studies (on the more challenging CALVIN domain) suggest that our temporal consistency objective enables more robust motor control. We believe that the temporal consistency objective can offer similar benefits for motor control across many manipulation and locomotion tasks, as these tasks often encompass highly structured action sequences. That said, another desirable property for intelligent agents is high-level cognitive reasoning abilities (planning, improvisation, etc). In this work we do not claim that our contributions facilitate aspects of high-level cognition. However we believe that our contributions can complement ideas in high-level cognition, and we leave the problem of integrating these ideas to future work.
> > > > > > > >
> > > > > > > > > for the baselines: any training on D_target alone is unfair as only the target task dataset is used. That said, It makes more sense after the ablation study when comparing to the "non-retrieval" study.
> > > > > > > >
> > > > > > > > Whenever applicable, we also include comparisons to methods that train on the prior dataset, such as BC-RNN (FT), FIST, and IQL (UDS). Even when comparing to methods that have access to both the target task demonstrations and prior datasets, our method outperforms these approaches, often by a wide margin. As you mentioned we also include ablations of our method that do not incorporate prior data. Our ablations (No Retrieval and No Prior Data) show that the prior data has a significant role both in training the policy and the skills.
> > > > > > > >
> > > > > > > > > What is meant by "struggles to find appropriate goal observations to reach"?
> > > > > > > >
> > > > > > > > FIST uses a semi-parametric policy to set (sub)goals to reach, and uses an inverse dynamics model (specifically the VAE prior from the skill model) to reach those goals. There may be two potential reasons why FIST underperforms: (1) the inverse dynamics model does not reach goals that it is given well, (2) the semi-parametric policy is setting incorrect / suboptimal goals to reach. We believe it is likely that (2) is the more likely explanation for FIST underperforming, however, since we have not fully validated this hypothesis we have updated the manuscript to include both explanations.
> > > > > > > >
> > > > > > > > > Why does FIST perform better for no-microwave?
> > > > > > > >
> > > > > > > > One possible explanation for this result is that the underlying model has greater optimization difficulties on the Kitchen-All dataset due to the greater scope and diversity of this dataset. In fact, from our experiment training logs we see that the skill losses are higher for Kitchen-All than Kitchen-No Microwave. In particular we see higher losses for the VAE inverse dynamics prior, resulting in skills that have lower fidelity in completing subgoals for Kitchen-All. Note that our method uses the skill model in a different manner as FIST (we use the skill encoder to determine which skill the policy should execute while FIST uses the VAE prior), and may be affected by this issue in different ways. An alternative hypothesis is that it is more difficult to optimize the contrastive distance model in FIST with Kitchen-All due to the increased diversity of this dataset (please refer to the FIST paper [1] for more details on this distance model). Further investigation is needed to examine these hypotheses.
> > > > > > > >
> > > > > > > > [1] Hierarchical few-shot imitation with skill transition models, Hakhamaneshi et al., ICLR 2022
> > > > > > > >
> > > > > > > > > Could the authors provide a comment on why BC-RNN (FT) performs so poorly on hardware experiments?
> > > > > > > >
> > > > > > > > For BC-RNN (FT) we found that the agent often has difficulty in grasping objects, especially the tomato. The agent oftentimes grasps in completely the wrong location. We noticed that this was the case consistently across multiple checkpoints. One hypothesis for the suboptimal behavior of BC-RNN (FT) specifically is that the policy is biased during the pre-training stage to perform multi-modal behaviors in the prior dataset, and this bias negatively impacts learning the target task.
> > > > > > > >
> > > > > > > > > Small comments on paper:
> > > > > > > >
> > > > > > > > Thank you for pointing out these details. In our updated manuscript we have addressed grammar and readability issues.

---

> > ### Comment · Reviewer_bYiA · 2022-08-26
> > **Thanks for the responses.**
> >
> > Thanks for the responses. I urge the authors to try to revise the paper and supplementary documents to reflect these responses. Most readers will only see the paper, not even the supplementary document or this openreview.net entry. Assume the reviewers are typical readers, and other readers will have the same questions.
> >
> > I have one final comment. At one point in the responses you effectively say the building blocks of behavior should be semantically meaningful. I disagree.

---

> > > ### Author Response · Authors · 2022-08-28
> > > **Thank you, please refer to our updated pdf**
> > >
> > > Thank you very much for your prompt feedback. We have taken care to incorporate as much of our openreview discussions as possible into the paper. Please note that due to space constraints it is challenging to include all of our discussions into the main text. However, we have added numerous references to specific appendix sections so that readers are fully aware of our additional experiments, implementation details, tasks and datasets, and hyperparameter details. You will also find that we have incorporated the majority of our rebuttal responses into the paper. Please refer to the post titled “Author Response to all Reviewers and Meta-reviewers” for our updated pdf.
> > >
> > > We are also working on building an interactive website with videos of our method and baselines on all of tasks. We will make an explicit link in the main text so that readers are aware of the website. We hope that the website will make the videos more accessible and help readers gain a better understanding of our method.
> > >
> > > Regarding your last point, upon reflection we believe that the notion of “semantically meaningful” skills is underspecified and can elicit different interpretations. For the sake of clarity we have thus removed references to “semantically meaningful” skills in our paper.

---

### Author Response · Authors · 2022-08-26
**Author Response to all Reviewers and Meta-reviewers**

**Comment:**

Dear reviewers and meta-reviewers,

We would wholeheartedly like to thank everyone for taking the time to review our work and providing constructive feedback. We received a number of great suggestions for strengthening our work and have accordingly made changes to our paper to incorporate these suggestions. We summarize these changes as follows:
* New real world task and new ablation result in real world
* Qualitative comparisons comparing retrieved sub-trajectories with random sub-trajectories, and without temporal consistency term
* Additional ablations on the target dataset size and retrieval dataset size and quality
* Retrieving sub-trajectories based on KL-divergence-based distance metric
* Additional discussion on prior work in skill-based imitation learning
* Updated IQL (UDS) result for Franka Kitchen after additional hyperparameter tuning
* Improved clarity throughout. Expanded discussion of method and low-level implementation details. Added references to figures, sections, and algorithm pseudocode. Resolved inconsistencies in terminology, notation, and grammar.

We have attached the updated manuscript and supplementary video materials. We hope that these changes have led to a noticeable improvement in our work. Please do not hesitate to let us know your feedback and any additional questions that you may have. Thank you!


**Zip File:**

/attachment/1d4ffed20d1c89ab8f8678f32861407aba76576e.zip

---

### Meta-Review · Area_Chair_oPFr · 2022-08-14

**Recommendation:** Accept (Poster)
**Confidence:** 4

**Metareview:**

Phase 1:

Strengths:
The submission targets a relevant problem and is written clearly and intuitive in its argument. The proposed method effectively combines known techniques with convincing empirical results and useful ablations which demonstrate the contributions of individual components.
The real world experiments find particular appreciation among reviewers.

Weaknesses:
While the presented experiments are helpful, multiple reviewers ask for further ablations to improve understanding (distance in embedding space wrt temporal aspects, impact of different data sizes for retrieval, comparison of resulting behaviors to retrieval dataset).
Further criticism applies to the limits of the used manipulation tasks and reviewers propose different paths to extend them. Finally, additional clarifications would also help to connect the method section to existing prior work.

Phase 2:

The feedback has been generally positive with some criticism. The submission targets a relevant problem and is clearly written and intuitive, convincing empirical results, real world experiments and useful ablations. Multiple reviewers ask for further ablations and clarifications to improve the understanding. I agree with the reviewers and recommend acceptance. Please take the remaining points from the review process seriously and follow up with improvements on open points and promised changes.

**Best Paper Nomination:**

No